# Exploring Large Action Sets with Hyperspherical Embeddings using von Mises-Fisher Sampling

## Abstract

This paper introduces von Mises-Fisher exploration (vMF-exp), a scalable method for exploring large action sets in reinforcement learning problems where hyperspherical embedding vectors represent actions. vMF-exp involves initially sampling a state embedding representation using a von Mises-Fisher distribution, then exploring this representation's nearest neighbors, which scales to virtually unlimited numbers of candidate actions. We show that, under theoretical assumptions, vMF-exp asymptotically maintains the same probability of exploring each action as Boltzmann Exploration (B-exp), a popular alternative that, nonetheless, suffers from scalability issues as it requires computing softmax values for each action. Consequently, vMF-exp serves as a scalable alternative to B-exp for exploring large action sets with hyperspherical embeddings. In the final part of this paper, we further validate the empirical relevance of vMF-exp by discussing its successful deployment at scale on a music streaming service. On this service, vMF-exp has been employed for months to recommend playlists inspired by initial songs to millions of users, from millions of possible actions for each playlist.

## 1 Introduction

Exploration is a fundamental component of the reinforcement learning (RL) paradigm (Amin et al., 2021; McFarlane, 2018; Sutton and Barto, 2018). It allows RL agents to gather valuable information about their environment and identify optimal actions that maximize rewards (Amin et al., 2021; Chiappa et al., 2023; Dulac-Arnold et al., 2015; Jin et al., 2020; Ladosz et al., 2022; McFarlane, 2018; Reynolds, 2002; Slivkins et al., 2019; Sutton and Barto, 2018; Tang et al., 2017). However, as the set of actions to explore grows larger, the exploration process becomes increasingly challenging. Indeed, large action sets can lead to higher computational costs, longer learning times, and the risk of inadequate exploration and suboptimal policy development (Amin et al., 2021; Chen et al., 2021; Dulac-Arnold et al., 2015; Lillicrap et al., 2016; Sutton and Barto, 2018; Tomasi et al., 2023).

As an illustration, consider a recommender system on a music streaming service like Apple Music or Spotify, curating playlists of songs "inspired by" an initial selection to help users discover music (Bendada et al., 2023a). In practice, these services often generate such playlists all at once, using efficient nearest neighbor search systems (Johnson et al., 2019; Li et al., 2019) to retrieve songs most similar to the initial one, in a song embedding vector space learned using collaborative filtering or content-based methods (Bendada et al., 2023a;b; Bontempelli et al., 2022; Jacobson et al., 2016; Schedl et al., 2018; Zamani et al., 2019). Alternatively, one could formalize this task as an RL problem (Tomasi et al., 2023), where the recommender system (i.e., the agent) would adaptively select the next song to recommend (i.e., the next action) based on user feedback on previously recommended songs (i.e., the rewards, such as likes or skips). Using an RL approach instead of generating the playlist at once would have the advantage of dynamically learning from user feedback to identify the best recommendations (Afsar et al., 2022; Tomasi et al., 2023). However, music streaming services offer access to large catalogs with millions of songs (Bendada et al., 2020; Jacobson et al., 2016; Schedl et al., 2018). Therefore, the agent would need to consider millions of possible actions for exploration, increasing the complexity of this task.

In particular, Boltzmann Exploration (B-exp) (Cesa-Bianchi et al., 2017; Sutton and Barto, 2018), a popular exploration strategy sampling actions to explore based on embedding similarities, would become practically intractable as it would require computing softmax values over millions of elements (see Section 2). Furthermore, in large action sets, many actions are often irrelevant; in our example, most songs would constitute poor recommendations (Tomasi et al., 2023). Therefore, random exploration methods like $\varepsilon$-greedy (Dann et al., 2022; Sutton and Barto, 2018), although more efficient than B-exp, would also be unsuitable for production use. Since these methods ignore song similarities, each song, including inappropriate ones, would have an equal chance of being selected for exploration. This could result in negative user feedback and a poor perception of the service (Tomasi et al., 2023). Lastly, deterministic exploration strategies would also be ineffective. Systems serving millions of users often rely on batch RL (Lange et al., 2012) since updating models after every trajectory is impractical. Batch RL, unlike on-policy learning, requires exploring actions non-deterministically given a state, and deterministic exploration would result in redundant trajectories and slow convergence (Bendada et al., 2020).

In summary, exploration remains challenging in RL problems characterized by large action sets and where accounting for embedding similarities is crucial, like our recommendation example. Overall, although a growing body of scientific research has been dedicated to adapting RL models for recommendation (see, e.g., the survey by Afsar et al. (2022)), evidence of RL adoption in commercial recommender systems exists but remains limited (Chen et al., 2019; 2021; 2022; Tomasi et al., 2023). The few existing solutions typically settle for a workaround by using a truncated version of B-exp (TB-exp). In TB-exp, a small subset of candidate actions is first selected, e.g., using approximate nearest neighbor search (a framework sometimes referred to as the Wolpertinger architecture (Dulac-Arnold et al., 2015)). Softmax values are then computed among those candidates only (Chen et al., 2019; 2021; 2022). YouTube, for instance, employs this technique for video recommendation (Chen et al., 2019). TB-exp allows for exploration in the close embedding neighborhood of a given state; however, it restricts the number of candidate actions based on technical considerations rather than optimal convergence properties. Although exploring beyond this restricted neighborhood might be beneficial, finding the best way to do so in large-scale settings remains an open research question.

In this paper, we propose to address this important question. Our work focuses on the specific setting where actions are represented by embedding vectors of dimension $d \geq 2$ with unit norm, i.e., embedding vectors lying on the $d$-dimensional unit hypersphere. As detailed in Section 2, this setting aligns with many real-world recommender system applications. Our contributions are as follows:

- We introduce von Mises-Fisher exploration (vMF-exp), a scalable method for exploring large sets of actions represented by hyperspherical embedding vectors. vMF-exp involves initially sampling a state embedding vector using a von Mises-Fisher distribution (Fisher, 1953), then exploring this representation's nearest neighbors. Our proposed strategy scales to millions of candidate actions and, unlike TB-exp, does not restrict exploration to a specific neighborhood.

- We provide a comprehensive analysis of vMF-exp, demonstrating that, under certain theoretical assumptions, it asymptotically maintains the same probability of exploring each action as the popular B-exp method, while overcoming its scalability issues. Consequently, vMF-exp serves as a scalable alternative to B-exp for effectively exploring large action sets.

- While our analysis remains general, we also offer a real-world example of a vMF-exp usage. We describe how, in 2024, we have deployed vMF-exp at scale on the music streaming service XXX[1] to recommend "Mixes inspired by" playlists. This application, backed by successful A/B tests on millions of users, confirms the empirical relevance of vMF-exp.

- We release a Python implementation of vMF-exp on GitHub to encourage its future use.

The remainder of this paper is organized as follows. We introduce our problem more formally in Section 2. We propose the vMF-exp method in Section 3. We present our theoretical analysis in Section 4, we discuss our experiments on XXX in Section 5, and we conclude in Section 6.

---

[1]We omit the name of this music streaming service to preserve anonymity during the review phase.

## 2 PRELIMINARIES

### 2.1 PROBLEM FORMULATION

**Notation** In this paper, we consider an RL agent sequentially selecting actions within a set $\mathcal{I}_n = \{1, 2, \ldots, n\}$ of $n \in \mathbb{N}^\star$ actions. Each action $i \in \mathcal{I}_n$ is represented by a distinct low-dimensional vectorial representation $X_i \in \mathbb{R}^d$, i.e., by an embedding vector or simply an embedding[2], for some fixed dimension $d \in \mathbb{N}$ with $d \geq 2$ and $d \ll n$. Additionally, we assume all vectors have a unit Euclidean norm, i.e., $\|X_i\|_2 = 1, \forall i \in \mathcal{I}_n$. They form a set of embeddings noted $\mathcal{X}_n = \{X_i, 1 \leq i \leq n\} \in (\mathcal{S}^{d-1})^n$, where $\mathcal{S}^{d-1}$ is the $d$-dimensional unit hypersphere (Fisher, 1953): $\mathcal{S}^{d-1} = \{x \in \mathbb{R}^d : \|X_i\|_2 = 1\}$.

We also assume the availability of an approximate nearest neighbor (ANN) (Johnson et al., 2019; Li et al., 2019) search engine. Using this engine, for any vector $V \in \mathcal{S}^{d-1}$, the nearest neighbor of $V$ among $\mathcal{X}_n$ in terms of inner product similarity (equal to the cosine similarity, for unit vectors (Tan et al., 2016)), called $X_{i_V^\star}$, can be retrieved in a sublinear time complexity with respect to $n$. Although ANN engines are parameterized based on a trade-off between efficiency and accuracy, we make the simplifying assumption that $X_{i_V^\star}$ is the actual nearest neighbor of $V$, which we later discuss in Section 4.3. Formally, $i_V^\star = \arg\max_{i \in \mathcal{I}_n} \langle V, X_i \rangle$.

Returning to the illustrative example of Section 1, $\mathcal{X}_n$ would represent embeddings associated with each song of the catalog $\mathcal{I}_n$ of the music streaming service. In this case, $n$ would be on the order of several millions (Bendada et al., 2020; Jacobson et al., 2016; Schedl et al., 2018). The RL agent would be the recommender system sequentially recommending these songs to users. Normalizing embeddings is a common practice in both academic and industrial recommender systems (Afchar et al., 2023; Bontempelli et al., 2022; Kim et al., 2023; Schedl et al., 2018) to mitigate popularity biases, as vector norms often encode popularity information on items (Afchar et al., 2023; Chen et al., 2023). Normalizing embeddings also prevents inner products from being unbounded, avoiding overflow and underflow numerical instabilities (LeCun et al., 2015).

At time $t$, the agent considers a state vector $V_t \in \mathcal{S}^{d-1}$, noted $V$ for brevity. It selects the next action in $\mathcal{I}_n$, whose relevance is evaluated by a reward provided by the environment. In our example, the agent would recommend the next song to continue the playlist, based on the previous song whose embedding $V$ acts as the current state. In this case, the reward might be based on user feedback, such as liking or skipping the song (Bontempelli et al., 2022). The agent may select $i_V^\star$, i.e., exploit $i_V^\star$ (Sutton and Barto, 2018). Alternatively, it may rely on an exploration strategy to select another $\mathcal{I}_n$ element. Formally, an exploration strategy $P$ is a policy function (Sutton and Barto, 2018) that, given $V$, selects each action $i \in \mathcal{I}_n$ with a probability $P(i \mid V) \in [0, 1]$.

**Objective** Our goal in this paper is to develop a suitable exploration strategy for our specific setting, where hyperspherical embedding vectors represent actions, and the number of actions can reach millions. Precisely, we aim to obtain an exploration scheme meeting the following properties:

- Scalability (**P1**): we consider an exploration scalable if the time required to sample actions given a vector $V$ is at most the time needed for the ANN engine to retrieve the nearest neighbor, which is typically achieved in a sublinear time complexity with respect to $n$. Scalability is a mandatory requirement for exploring large action sets with millions of elements.

- Unrestricted radius (**P2**): Radius($P \mid V$) is the number of actions with a non-zero probability of being explored given a state $V$. While exploring actions too far from $V$ might be suboptimal (e.g., resulting in poor recommendations), it is crucial that exploration is not restricted to a specific radius by construction. Such a restriction could prevent the agent from exploring relevant actions that lie beyond this radius. An unrestricted radius ensures that the exploration strategy remains flexible and capable of adapting to various contexts, allowing for the exploration of relevant actions regardless of their embedding position.

- Order preservation (**P3**): order is preserved when the probability of selecting the action $i$ given the state $V$ is a strictly increasing function of $\langle V, X_i \rangle$. More formally, order preservation requires $\forall (i, j) \in \mathcal{I}_n^2, \langle V, X_i \rangle > \langle V, X_j \rangle \implies P(i \mid V) > P(j \mid V)$.

---

[2]At this stage, we do not make assumptions regarding the specific methods or data used to learn these embedding vectors, nor the precise interpretation of proximity between vectors in the embedding space.

Order preservation implies that the exploration strategy properly leverages the information captured in the embedding vectors to assess the relevance of an action given a state.

## 2.2 LIMITATIONS OF EXISTING EXPLORATION STRATEGIES

Finding an exploration strategy that simultaneously meets these three properties is essential for effective exploration in RL problems with large action sets and embedding representations. Nonetheless, existing exploration strategies suffer from limitations that motivate our work in this paper.

**Random and $\varepsilon$-greedy Exploration**  The most straightforward example of an exploration strategy would be the random (uniform) policy, where $P_{\mathrm{rand}}(i \mid V) = \frac{1}{n}, \forall i \in \mathcal{I}_n$. A popular variant is the $\varepsilon$-greedy strategy (Sutton and Barto, 2018). With a probability $\varepsilon \in [0, 1]$, the agent would choose the next action uniformly at random. With a probability $1 - \varepsilon$, it would exploit the most relevant action based on its knowledge. Random and $\varepsilon$-greedy exploration strategies are scalable (**P1**), as elements of $\mathcal{I}_n$ can be uniformly sampled in $\mathcal{O}(1)$ time (Cormen et al., 2022). Additionally, they verify **P2**. Indeed, $\mathrm{Radius}(P_{\mathrm{rand}}|V) = n$ since every action can be selected. However, these strategies ignore embeddings at the sampling phase and do not achieve order preservation (**P3**). This is a significant limitation, reinforced by the fact that these policies have a maximal radius. As explained in Section 1, in large action sets, many actions are often irrelevant, e.g., most songs from the musical catalog would constitute poor recommendations given an initial state (Tomasi et al., 2023). Exploring each action/song with equal probability, including inappropriate ones, could result in negative user feedback and a poor perception of the service (Tomasi et al., 2023).

**Boltzmann Exploration**  To address the limitations of random exploration, one can sample actions according to their embedding similarity with $V$. The prevalent approach in RL is Boltzmann Exploration (B-exp) (Amin et al., 2021; Cesa-Bianchi et al., 2017; Chen et al., 2021; Sutton and Barto, 2018), which employs the Boltzmann distribution for action sampling:

$$\forall i \in \mathcal{I}_n, P_{\mathrm{B\text{-}exp}}(i \mid V, \mathcal{X}_n, \kappa) = \frac{\mathrm{e}^{\kappa \langle V \,,\, X_i \rangle}}{\sum_{j=1}^{n} \mathrm{e}^{\kappa \langle V \,,\, X_j \rangle}}, \tag{1}$$

where the hyperparameter $\kappa \in \mathbb{R}^+$ controls the entropy of the distribution. B-exp samples actions according to a strictly increasing function of their inner product similarity with $V$ for $\kappa > 0$, guaranteeing order preservation (**P3**). By carefully tuning $\kappa$, one can ensure that irrelevant actions are practically never selected while maintaining a non-zero probability of recommending actions with less than maximal similarity, thereby indirectly controlling the radius of the policy (**P2**). Unfortunately, B-exp does not satisfy **P1**, i.e., it is not scalable to large action sets. Indeed, evaluating Equation (1) requires explicitly computing the probability of sampling each individual action before actually sampling from them, which is prohibitively expensive for large values of $n$ (Chen et al., 2021). Note that, while we focus on B-exp in this section, these scalability concerns would remain valid for any other sampling distribution requiring explicitly computing similarities and probabilities for each of the $n$ actions (Amin et al., 2021).

**Truncated Boltzmann Exploration**  Due to these scalability concerns, previous work on RL with large and embedded action sets often settled for a workaround consisting in sampling actions from a truncated version of the Boltzmann distribution (or another distribution) (Chen et al., 2021). In this method, which we refer to as Truncated Boltzmann Exploration (TB-exp), a small number $m \ll n$ of candidate actions, usually around hundreds or thousands, is first retrieved using the ANN search engine, leading to a candidate action set $\mathcal{I}_m(V)$. The sampling step is subsequently performed only within $\mathcal{I}_m(V)$:

$$\forall i \in \mathcal{I}_m(V), P_{\mathrm{TB\text{-}exp},m}(i \mid V, \mathcal{X}_n, \kappa) = \frac{\mathrm{e}^{\kappa \langle V \,,\, X_i \rangle}}{\sum_{j \in \mathcal{I}_m(V)} \mathrm{e}^{\kappa \langle V \,,\, X_j \rangle}}. \tag{2}$$

TB-exp performs action selection in a time that depends on $m$ instead of $n$, and has been successfully deployed in production environments involving millions of actions (Chen et al., 2019; 2021; 2022). While it still satisfies **P3**, TB-exp also meets **P1** for small values of $m$. However, it no longer satisfies **P2**. This method restricts the radius, i.e., the number of candidate actions, based on technical considerations rather than exploration efficiency. This restriction can potentially hinder model convergence

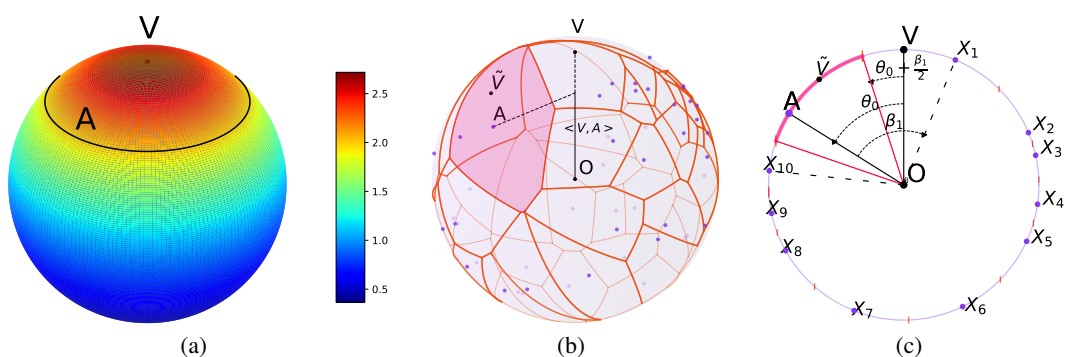

(a)  (b)  (c)

Figure 1: (a) PDF of a 3D vMF distribution. (b) vMF-exp explores the action $A$ when the sampled vector $\tilde{V}$ lies in $A$'s Voronoï cell, shown in red in 3D. (c) Same as (b) in 2D.

by neglecting the exploration of relevant actions beyond this fixed radius. In summary, the challenge of finding an exploration strategy that satisfies **P1**, **P2**, and **P3** simultaneously—in other words, an exploration scheme with properties similar to full Boltzmann exploration yet scalable—remains relatively open.

## 3 FROM BOLTZMANN TO VON MISES–FISHER (vMF) EXPLORATION

In this section, we present our solution for exploring large action sets with hyperspherical embeddings.

### 3.1 VON MISES–FISHER EXPLORATION

The inability of B-exp to scale arises from its need to compute all $n$ sampling probabilities explicitly. In this paper, we propose von Mises-Fisher Exploration (vMF-exp), an alternative exploration strategy that overcomes this constraint. Specifically, given an initial state vector $V$, vMF-exp consists in:

- Firstly, sampling a vector $\tilde{V}$ according to a vMF distribution (Fisher, 1953) centered on $V$.
- Secondly, selecting $\tilde{V}$s nearest neighbor action in the embedding space for exploration.

In directional statistics, the vMF distribution (Fisher, 1953) is a continuous vector probability distribution defined on the unit hypersphere $\mathcal{S}^{d-1}$. It has recently been used in RL to assess the uncertainty of gradient directions (Zhu et al., 2024). For all $\tilde{V} \in \mathcal{S}^{d-1}$, its probability density function (PDF) is:

$$f_{\text{vMF}}(\tilde{V} \mid \kappa, V, d) = C_d(\kappa)e^{\kappa\langle V, \tilde{V}\rangle}, \text{with } C_d(\kappa) = \frac{1}{\int_{\tilde{V}\in\mathcal{S}^{d-1}} e^{\kappa\langle V, \tilde{V}\rangle}\,\mathrm{d}\tilde{V}} = \frac{\kappa^{\frac{d}{2}-1}}{(2\pi)^{\frac{d}{2}}I_{\frac{d}{2}-1}(\kappa)} \quad (3)$$

with $\kappa \in \mathbb{R}^+$. The function $I_{\frac{d}{2}-1}$ designates the modified Bessel function of the first kind (Baricz, 2010) at order $d/2 - 1$. Figure 1(a) illustrates the PDF of a vMF distribution on the 3-dimensional unit sphere. For any $\tilde{V} \in \mathcal{S}^{d-1}$, $f_{\text{vMF}}(\tilde{V} \mid \kappa, V, d)$ is proportional to $e^{\kappa\langle V, \tilde{V}\rangle}$, which is reminiscent of the B-exp sampling probability of Equation (1). The hyperparameter $\kappa$ controls the entropy of the distribution. In particular, for $\kappa = 0$, the vMF distribution boils down to the uniform distribution on $\mathcal{S}^{d-1}$.

### 3.2 PROPERTIES

**P1** vMF-exp only requires sampling a $d$-dimensional vector instead of handling a discrete distribution with $n$ parameters, allowing $\tilde{V}$ to be sampled in constant time with respect to $n$. Therefore, vMF-exp is a scalable exploration strategy. Efficient sampling algorithms for vMF distributions have been extensively studied (Kang and Oh, 2024; Pinzón and Jung, 2023). As shown in the following sections, we successfully explored sets of millions of actions without scalability issues, using the Python vMF sampler from Pinzón and Jung (2023) for simulations in Section 4 and our own variant implementation for A/B tests in Section 5.

**P2**   The probability of sampling $i \in \mathcal{I}_n$ given $V$ for exploration is the probability that $X_i$ is the nearest neighbor of $\tilde{V}$ among $\mathcal{X}_n$ vectors, i.e., that $\tilde{V}$ lies in $\mathcal{S}_{\text{Voronoï}}(X_i \mid \mathcal{X}_n)$, the Voronoï cell of $X_i$ in the Voronoï tessellation of $\mathcal{S}^{d-1}$ defined by $\mathcal{X}_n$ (Du et al., 1999; 2010) (see Figures 1(b) and 1(c)). We have: $\mathcal{S}_{\text{Voronoï}}(X_i \mid \mathcal{X}_n) = \{\tilde{V} \in \mathcal{S}^{d-1}, \forall j \in \mathcal{I}_n, \langle \tilde{V} \ , \ X_i \rangle \geq \langle \tilde{V} \ , \ X_j \rangle\}$, and $\bigcup_{i \in \mathcal{I}_n} \mathcal{S}_{\text{Voronoï}}(X_i \mid \mathcal{X}_n) = \mathcal{S}^{d-1}$. Using this notation, the probability of sampling the action $i$ for exploration using vMF-exp is:

$$\forall i \in \mathcal{I}_n, P_{\text{vMF-exp}}(i \mid V, \mathcal{X}_n, \kappa) = \int_{\tilde{V} \in \mathcal{S}_{\text{Voronoï}}(X_i \mid \mathcal{X}_n)} f_{\text{vMF}}(\tilde{V} \mid \kappa, V, d) \, \mathrm{d}\tilde{V}, \tag{4}$$

which is always strictly positive. Thus, vMF-exp verifies the unrestricted radius property. Like B-exp, tuning $\kappa$ ensures that actions with low similarity have negligible sampling probabilities in practice.

**P3**   $P_{\text{vMF-exp}}(i \mid V, \mathcal{X}_n, \kappa)$ increases due to two factors. Firstly, the average $f_{\text{vMF}}(\tilde{V} \mid \kappa, V, d)$ value for $\tilde{V} \in \mathcal{S}_{\text{Voronoï}}(X_i \mid \mathcal{X}_n)$, which is correlated to $\langle X_i \ , \ V \rangle$ and contributes to order preservation. Secondly, the surface area of $\mathcal{S}_{\text{Voronoï}}(X_i \mid \mathcal{X}_n)$, measuring how dissimilar $X_i$ is from other $\mathcal{X}_n$ elements. Actions embedded in a low-density subspace of $\mathcal{S}^{d-1}$ will have an expanded Voronoï cell and may be selected more often than actions closer to $V$ but located in a high-density subspace. Hence, vMF-exp favors actions that are both similar to $V$ and dissimilar to other actions, and order preservation depends on the $\mathcal{X}_n$ distribution. Section 4 will focus on a setting where B-exp and vMF-exp asymptotically share similar probabilities. Consequently, vMF-exp, like B-exp, will verify order preservation (**P3**). In conclusion, in this setting, vMF-exp will verify **P1**, **P2**, and **P3** simultaneously.

## 4   THEORETICAL COMPARISON OF VMF-EXP AND B-EXP

We now provide a mathematical comparison of vMF-exp and B-exp. We focus on the theoretical setting presented in Section 4.1. We show that, in this setting, vMF-exp maintains the same probability of exploring each action as B-exp, while overcoming its scalability issues. As noted above, this implies that vMF-exp verifies **P1**, **P2**, and **P3** simultaneously and, therefore, acts as a scalable alternative to the popular but unscalable B-exp for exploring large action sets with hyperspherical embeddings.

### 4.1   SETTING AND ASSUMPTIONS

We focus on the setting where embeddings are independent and identically distributed (i.i.d.) and follow a uniform distribution on the unit hypersphere, i.e., $\mathcal{X}_n \sim \mathcal{U}(\mathcal{S}^{d-1})$. For convenience in our proofs, we consider the action set to be the union of $\mathcal{I}_n$, the set of $n$ actions, and another action $a$ with a known embedding $A \in \mathcal{S}^{d-1}$. The resulting entire action set $\mathcal{I}_{n+1}$ and embedding set $\mathcal{X}_{n+1}$ are defined as $\mathcal{I}_{n+1} = \mathcal{I}_n \cup \{a\}$ and $\mathcal{X}_{n+1} = \mathcal{X}_n \cup \{A\}$. In this section, we are interested in the probability of each exploration scheme, B-exp and vMF-exp, to sample $a$ among all actions of $\mathcal{I}_{n+1}$ given a state embedding vector $V \in \mathcal{S}^{d-1}$. These probabilities are defined respectively as:

$$P_{\text{B-exp}}(a \mid n, d, V, \kappa) = \mathbb{E}_{\mathcal{X}_n \sim \mathcal{U}(\mathcal{S}^{d-1})}\Big[P_{\text{B-exp}}(a \mid V, \mathcal{X}_{n+1}, \kappa)\Big], \tag{5}$$

$$P_{\text{vMF-exp}}(a \mid n, d, V, \kappa) = \mathbb{E}_{\mathcal{X}_n \sim \mathcal{U}(\mathcal{S}^{d-1})}\Big[P_{\text{vMF-exp}}(a \mid V, \mathcal{X}_{n+1}, \kappa)\Big]. \tag{6}$$

### 4.2   RESULTS

We now present and discuss our main theoretical results. For brevity, we report all intermediary lemmas and mathematical proofs in the Appendices A to D of this paper. Our first and most general result links the asymptotic behavior of B-exp and vMF-exp as the action set grows.

**Proposition 4.1.** *In the setting of Section 4.1, we have:*

$$\lim_{n \to +\infty} \frac{P_{\text{B-exp}}(a \mid n, d, V, \kappa)}{P_{\text{vMF-exp}}(a \mid n, d, V, \kappa)} = 1. \tag{7}$$

Proposition 4.1 states that, for large values of $n$, the probability of selecting $a$ for exploration is asymptotically the same using either B-exp or vMF-exp. This result follows from the respective

asymptotic characterizations of $P_{\text{B-exp}}$ and $P_{\text{vMF-exp}}$, detailed below. Importantly, it implies that, for large values of $n$, vMF-exp shares the same properties as B-exp (**P2**, **P3**), including order preservation. However, as noted in Section 3, vMF-exp offers greater scalability since its implementation only requires sampling a vector of a fixed size $d$, an operation independent of the number of actions $n$ (**P1**). Next, we give a common approximate expression for both methods, defined as $P_0(a \mid n, d, V, \kappa) = \frac{f_{\text{vMF}}(A|V,\kappa)\mathcal{A}(\mathcal{S}^{d-1})}{n}$, with $\mathcal{A}(\mathcal{S}^{d-1})$ denoting the surface area of the hypersphere $\mathcal{S}^{d-1}$, and describe the rate at which this asymptotic behavior is reached as $n$ grows.

**Proposition 4.2.** *In the setting of Section 4.1, we have:*

$$P_{\text{B-exp}}(a \mid n, d, V, \kappa) = P_0(a \mid n, d, V, \kappa) + o(\frac{1}{n\sqrt{n}}). \tag{8}$$

**Proposition 4.3.** *In the setting of Section 4.1, we have:*

$$P_{\text{vMF-exp}}(a \mid n, d, V, \kappa) = P_0(a \mid n, d, V, \kappa) + \begin{cases} \mathcal{O}(\frac{1}{n^2}) & \text{if } d = 2, \\ \mathcal{O}(\frac{1}{n^{1+\frac{2}{d-1}}}) & \text{if } d > 2. \end{cases} \tag{9}$$

In essence, when $n$ is large, the probability of sampling the action $a$ can be approximated by the PDF of the vMF distribution evaluated at $A$ multiplied by the average surface area of the Voronoï cell of $A$, for both exploration methods. As $n$ grows, this Voronoï cell shrinks until $f_{\text{vMF}}$ becomes nearly constant across its entire surface. Figure 2(f) illustrates this interpretation.

However, the rate at which both exploration methods reach their asymptotic behavior differs. The rate at which the Voronoi cell shrinks depends on the dimension of the hypersphere, which explains why the second term of Equation (9) depends on $d$. Note that this is not the case for B-exp. Consequently, for large values of $d$, one may require a higher number of actions $n$ before the asymptotic behavior of Equation (7) is observed. For this reason, it is useful to obtain a more precise approximation of $P_{\text{vMF-exp}}(a \mid n, d, V, \kappa)$ when $d$ increases, which we provide in the next section.

### 4.3 DISCUSSION

**High Dimension**   Following the above discussion, Proposition 4.4 offers a more precise expression of $P_{\text{vMF-exp}}(a \mid n, V, \kappa)$ to use when $d$ increases (roughly, $d \geq 20$ in our experiments). It is obtained by studying the first two terms of the Taylor expansion (Abramowitz and Stegun, 1948) of $f_{\text{vMF}}$ near $A$, instead of only the zero-order term, the second term becoming more significant when $d$ increases. Despite its apparent complexity, it can be interpreted simply. The negative sign before $\langle V, A \rangle$, indicates that, when $A$ is similar to $V$, it is sampled less often than with B-exp for the same $\kappa$ and $d$ values. Conversely, when $A$ is on the opposite side of the hypersphere, the term contributes positively to $P_{\text{vMF-exp}}(a \mid n, V, \kappa)$. To summarize, for larger $d$ values, vMF-exp is expected to explore more than B-exp with the same $\kappa$.

**Proposition 4.4.** *Let $B : (z_1, z_2) \mapsto \int_0^1 t^{z_1-1}(1-t)^{z_2-1}\,dt$ denote the Beta function, and $\Gamma : z \mapsto \int_0^\infty t^{z-1}e^{-t}\,dt$ denote the Gamma function (Abramowitz and Stegun, 1948). In the setting of Section 4.1 with $d \geq 3$, we have:*

$$P_{\text{vMF-exp}}(a \mid n, V, \kappa) = P_1(a \mid n, V, \kappa) + \mathcal{O}(\frac{1}{n^{\frac{2}{d-1}}}), \text{with:} \tag{10}$$

$$P_1(a \mid n, V, \kappa) = P_0(a \mid n, V, \kappa) - \frac{f_{\text{vMF}}(A \mid V, \kappa)\mathcal{A}(\mathcal{S}^{d-1})}{n} \frac{\kappa\langle V, A\rangle\Gamma(\frac{d+1}{d-1})}{2} \left( \frac{(d-1)B(\frac{1}{2}, \frac{d-1}{2})}{n} \right)^{\frac{2}{d-1}}.$$

**The case $d = 2$**   In 2 dimensions, Voronoi cells are arcs of a circle and are delimited by the perpendicular bisectors of two neighboring points, as shown in Figure 1(c). Interestingly, in this specific case, $P_{\text{vMF-exp}}(a \mid n, d = 2, V, \kappa)$ can be computed using geometric arguments. We report a comprehensive analysis in Appendix B, confirming that, when $d = 2$, vMF-exp approaches its asymptotic behavior faster than B-exp, as indicated by the $\mathcal{O}(\frac{1}{n^2})$ term in Proposition 4.3.

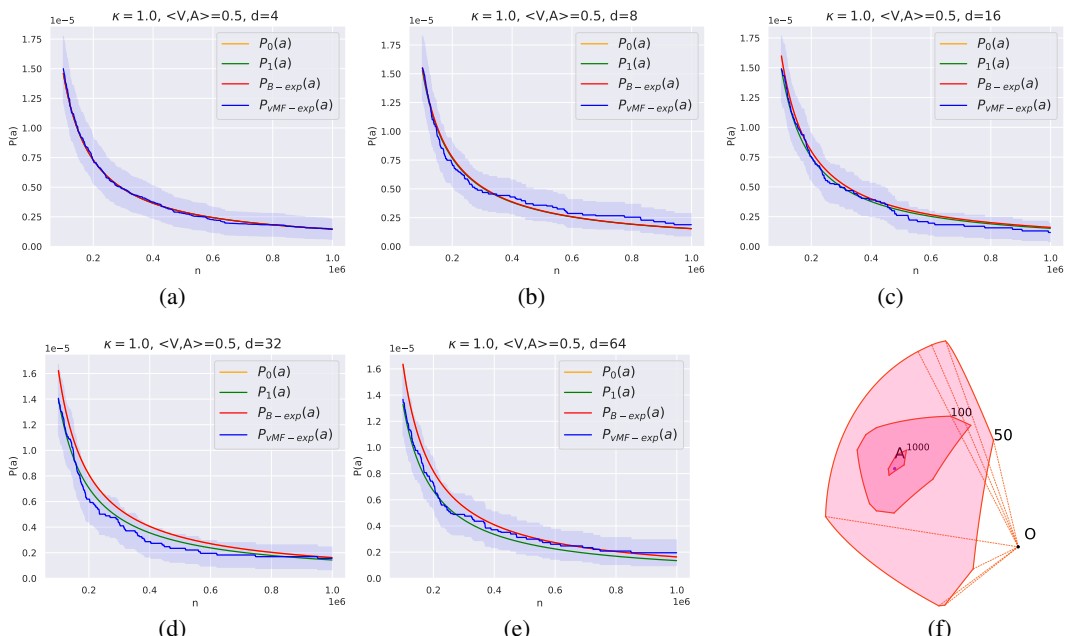

Figure 2: (a) to (e): Simulations of Section 4.3. (f) 3D Voronoï cell of $A$ for $n \in \{50, 100, 1000\}$.

**Validation Using Monte Carlo Simulations** Using the Python sampler of Pinzón and Jung (2023), we repeatedly sampled vectors $\mathcal{X}_n \sim \mathcal{U}(S^{d-1})$ and $\tilde{V} \sim \text{vMF}(V, \kappa)$, for various $d$, $\kappa$, and $\langle V , A \rangle$. Figure 2 reports, for $\kappa = 1.0$, $\langle V , A \rangle = 0.5$ and growing values of $d$, the $P_{\text{vMF-exp}}(a)$ sampling probability depending on the number of actions $n$, as well as $P_{\text{B-exp}}(a)$ with similar parameters and our approximations $P_0(a)$ and $P_1(a)$. We repeated all experiments 8 million times and reported 95% intervals. Our results are consistent with our theoretical findings. Firstly, in line with Proposition 4.2, $P_{\text{B-exp}}(a)$ and $P_0(a)$ are indistinguishable for this range of $n$ values. Secondly, for small $d$ values (Figures 2(a), 2(b), 2(c)), $P_{\text{vMF-exp}}$ is also tightly aligned with $P_{\text{B-exp}}(a)$ and $P_0(a)$, consistently with Proposition 4.1 and 4.3. Note that the y-axis is on a 1e-5 scale; hence, probabilities are extremely close. Thirdly, when $d \geq 16$ (Figures 2(d), 2(e)), $P_1(a)$ becomes more distinguishable from $P_0(a)$ and constitutes a better approximation of $P_{\text{vMF-exp}}(a)$ than $P_0(a)$, as per Proposition 4.4. Lastly, since $\langle V , A \rangle > 0$, Proposition 4.4 predicts that $P_{\text{B-exp}}(a) \geq P_{\text{vMF-exp}}(a)$ for large $d$, which our experiments confirm. We provide comparable simulations with other $(d, \kappa, \langle V , A \rangle)$ combinations in Appendix F. Our code is available online: `https://github.com/removed/for-anonymity`.

**Link with Thompson Sampling** One might draw interesting similarities between vMF-exp and bandit arm exploration using Thompson Sampling (Chapelle and Li, 2011). Appendix E compares the two approaches.

**Limitations and Future Work** While we believe our study offers valuable insights into vMF-exp, several limitations must be acknowledged. Most notably, our theoretical guarantees are currently restricted to the setting of Section 4.1 where embeddings are i.i.d. and uniform vectors. Although, in practice, vMF-exp can be used with hyperspherical embeddings from other distributions, we do not yet provide guarantees in these cases. For instance, studying vMF-exp in clustered embedding settings, as is sometimes the case with music recommendation embeddings (Afchar et al., 2023) (where clusters can, e.g., summarize music genres (Salha-Galvan et al., 2022)), could be insightful. We believe that this future work should benefit from the methods used to derive the non-trivial demonstration for the uniform distribution case. Section 5 will demonstrate the practical value of vMF-exp on song embeddings that do not explicitly comply with Section 4.1, but further mathematical investigation would be warranted. In future work, we will also study the second-order term of Proposition 4.4, which could be relevant for large values of $\kappa$, and the impact of errors from the ANN engine. While we assumed this engine returns exact neighbors, this may not hold for very large action sets (Johnson et al., 2019) and, intuitively, could cause minor exploration perturbations.

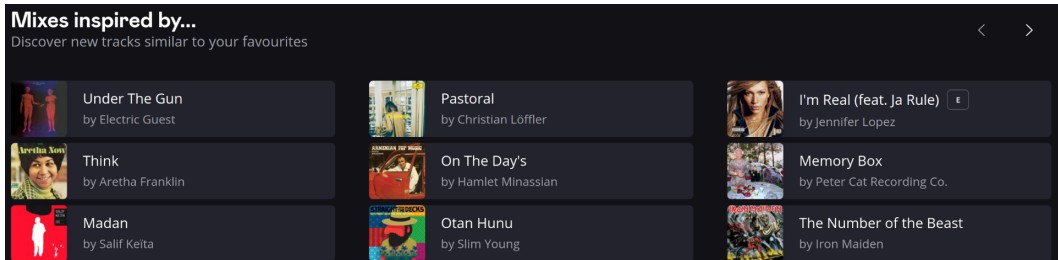

Figure 3: Interface of the "Mixes inspired by" recommender system on XXX. To preserve anonymity, we have removed some elements from the figure, such as the logo and the top/bottom of the website.

## 5 APPLICATION TO LARGE-SCALE MUSIC RECOMMENDATION

Our analysis of vMF-exp in Section 4 was intentionally general, as the method can be applied to various problem settings. In this Section 5, we showcase a real-world application of vMF-exp.

### 5.1 EXPERIMENTAL SETTING

We consider the "Mixes inspired by" feature of the global music streaming service XXX. This recommender system is deployed at scale and available on the homepage of this service. As shown in Figure 3, it displays a personalized shortlist of songs, selected from those previously liked by each user. A click on a song generates a playlist of 40 songs "inspired by" the initial one, with the aim of helping users discover new music within a catalog including several millions of recommendable songs.

To generate playlists, XXX leverages a collaborative filtering model (Koren and Bell, 2015). This model learns unit norm song embedding representations of dimension $d = 128$ by factorizing a mutual information matrix based on song co-occurrences in various listening contexts, using singular value decomposition (SVD) (Banerjee and Roy, 2014). Inner product proximity in the resulting embedding space aims to reflect user preferences. When a user selects an initial song, the model retrieves its embedding, then (approximately) identifies its neighbors in the embedding space using the efficient Faiss library (Johnson et al., 2019) for ANN. Currently, XXX generates the entire playlist at once in production. The service is considering RL approaches to, instead, recommend songs one by one while adapting to user feedback on previous songs of the playlist (likes, skips, etc.). However, as explained in Section 1, adopting such approaches would require exploring millions of possible actions/songs, significantly increasing the complexity of this task. In this section, we continue generating "Mixes inspired by" playlists all at once, but take a step towards RL by comparing three methods for exploring large action sets of millions of songs:

- vMF-exp: we use the embedding of the user's selected song as the initial state $V$. We sample a random state embedding $\tilde{V}$ according to the vMF distribution, using the estimator of Banerjee et al. (2005) to tune $\kappa$ (see Equation (4) of Sra (2012)). Finally, we recommend the 40 nearest neighbors of $\tilde{V}$ in the embedding space according to the ANN engine.

- TB-exp: comparing vMF-exp to full B-exp is practically intractable at this scale. We compare vMF-exp to TB-exp with a similar $\kappa$. We first retrieve the $m = 500$ nearest neighbors of the initial song in the embedding space, according to the ANN engine. Then, we generate the playlist by sampling 40 songs from these 500 using a truncated Boltzmann distribution.

- Reference: we also compare vMF-exp to a baseline that retrieves the 500 nearest neighbors of the initial song using ANN, then shuffles them randomly to generate a playlist of 40 songs.

In early 2024, we conducted an online A/B test on XXX to compare these exploration strategies in real conditions. The test involved millions of users worldwide, randomly split and unaware of the test.

### 5.2 RESULTS

Firstly, it is important to highlight that we were able to successfully deploy vMF-exp in XXX's production environment, achieving a sampling latency of just a few milliseconds, comparable to the

other methods. This industrial deployment on a service used by millions of users on a daily basis confirms the claimed scalability of vMF-exp and its practical relevance for large-scale applications.

Using vMF-exp or TB-exp for exploration improved the daily number of recommended songs "liked" by users through "Mixes inspired by" (liking a song adds it to their list of favorites), compared to the reference baseline. For confidentiality, we do not report exact numbers of likes or users in each cohort, but present relative rates with respect to the reference. On average, users exposed to vMF-exp or TB-exp added 11% more recommended songs to their playlists than the reference cohort. These differences were statistically significant at the 1% level (p-value < 0.01). No apparent differences were observed between vMF-exp and TB-exp, showing that vMF-exp is competitive with TB-exp.

In addition, vMF-exp, which does not suffer from the restricted radius of TB-exp, recommended more diverse playlists. We measured the average Jaccard similarity (Tan et al., 2016) of playlists generated from the same initial selection, to assess how similar the songs sampled from the same state embedding were, for each method. Results reveal that TB-exp had an average Jaccard similarity 35% higher (less diverse playlists) than vMF-exp, a statistically significant difference at the 1% level (p-value < 0.01). Therefore, vMF-exp allowed for a more substantial exploration, without compromising performance.

At the time of writing, XXX continues to use vMF-exp for "Mixes inspired by" recommendations. Playlists are still generated at once, but our work equips this service with an effective strategy to explore their large and embedded action set of millions of songs. This opens interesting avenues for further investigation of RL for recommendation. In the near future, XXX will launch tests involving actor-critic RL models (Konda and Tsitsiklis, 1999; Sutton and Barto, 2018) to explore and generate songs sequentially based on user feedback.

## 6 CONCLUSION

In conclusion, the primary contribution of this article is the development of vMF-exp, a scalable method for exploring large action sets in RL problems where hyperspherical embedding vectors represent actions. We have shown that, under theoretical conditions, vMF-exp asymptotically maintains the same probability of exploring each action as the popular B-exp method while overcoming its scalability issues. Additionally, unlike the TB-exp workaround, which restricts exploration to a specific neighborhood, vMF-exp allows for unrestricted exploration. This makes vMF-exp a valuable tool for RL researchers and practitioners aiming to explore large action sets with hyperspherical embeddings, offering a suitable alternative to both B-exp and TB-exp. We have also discussed the limitations of our work, suggesting directions for future research. While our analysis has been general, the final part of this article has also provided a real-world application of vMF-exp. Specifically, we have successfully deployed vMF-exp on the music streaming service XXX, where it has been used for months to better explore songs to recommend to millions of users. This application highlights the practical relevance of our work and will facilitate future RL research and large-scale experiments on XXX.

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

## APPENDIX

This appendix provides detailed proofs and discussions for all theoretical results presented in the "*Exploring Large Action Sets with Hyperspherical Embeddings using von Mises-Fisher Sampling*" article, along with additional figures and experiments on public datasets.

## A  ASYMPTOTIC BEHAVIOR OF BOLTZMANN EXPLORATION (PROOF OF PROPOSITION 4.2)

We begin with the proof of Proposition 4.2 claiming that, in the setting of Section 4.1, we have:

$$P_{\text{B-exp}}(a \mid n, d, V, \kappa) = \underbrace{\frac{f_{\text{vMF}}(A \mid V, \kappa)\mathcal{A}(\mathcal{S}^{d-1})}{n}}_{\text{denoted } P_0(a \mid n, d, V, \kappa)} + o(\frac{1}{n\sqrt{n}}), \tag{11}$$

with $f_{\text{vMF}}$ the probability density function (PDF) of the von Mises-Fisher (vMF) (Fisher, 1953) distribution:

$$\forall A \in \mathcal{S}^{d-1}, f_{\text{vMF}}(A \mid V, \kappa) = C_d(\kappa)e^{\kappa\langle V , A\rangle}, \tag{12}$$

where $\mathcal{A}(\mathcal{S}^{d-1})$ is the surface area of $\mathcal{S}^{d-1}$, the $d$-dimensional unit hypersphere, and $C_d(\kappa)$ is the normalizing constant.

*Proof.* By definition,

$$\begin{aligned}
P_{\text{B-exp}}(a \mid n, d, V, \kappa) &= \mathbb{E}_{\mathcal{X}_n \sim \mathcal{U}(\mathcal{S}^{d-1})}\left[\frac{e^{\kappa\langle V , A\rangle}}{e^{\kappa\langle V , A\rangle} + \sum_{i=1}^{n} e^{\kappa\langle V , X_i\rangle}}\right] \\
&= \frac{e^{\kappa\langle V , A\rangle}}{n} \mathbb{E}_{\mathcal{X}_n \sim \mathcal{U}(\mathcal{S}^{d-1})}\left[\frac{1}{\frac{e^{\kappa\langle V , A\rangle}}{n} + \sum_{i=1}^{n} \frac{e^{\kappa\langle V , X_i\rangle}}{n}}\right] \\
&= \frac{e^{\kappa\langle V , A\rangle}}{n} \mathbb{E}_{\mathcal{X}_n \sim \mathcal{U}(\mathcal{S}^{d-1})}\left[\frac{1}{D_n}\right].
\end{aligned} \tag{13}$$

We use $D_n$ to denote the denominator of the expression inside the above expectation. $D_n$ is the empirical average of $n$ independent and identically distributed (i.i.d.) random variables (plus a constant). Therefore, by applying the *Central Limit Theorem (CLT)* (Fischer, 2011), we know that as $n$ grows it will be asymptotically distributed according to a Normal distribution with the following expectation:

$$\begin{aligned}
\mathbb{E}_{\mathcal{X}_n \sim \mathcal{U}(\mathcal{S}^{d-1})}[D_n] &= \mathbb{E}_{\mathcal{X}_n \sim \mathcal{U}(\mathcal{S}^{d-1})}\left[\frac{e^{\kappa\langle V , A\rangle}}{n} + \sum_{i=1}^{n} \frac{e^{\kappa\langle V , X_i\rangle}}{n}\right] \\
&= \frac{e^{\kappa\langle V , A\rangle}}{n} + \mathbb{E}_{X \sim \mathcal{U}(\mathcal{S}^{d-1})}\left[e^{\kappa\langle V , X\rangle}\right].
\end{aligned} \tag{14}$$

Moreover, we have:

$$\begin{aligned}
\mathbb{E}_{X \sim \mathcal{U}(\mathcal{S}^{d-1})}\left[e^{\kappa\langle V , X\rangle}\right] &= \int_{X \in \mathcal{S}^{d-1}} \frac{e^{\kappa\langle V , X\rangle}}{\mathcal{A}(\mathcal{S}^{d-1})} \, dX \\
&= \frac{1}{\mathcal{A}(\mathcal{S}^{d-1})C_d(\kappa)},
\end{aligned} \tag{15}$$

using the fact that $C_d(\kappa)$ is the normalizing constant of a vMF distribution, ensuring that its PDF (Equation (12)) sums to 1 when integrated on the unit hypersphere.

Let us define $\sigma = \text{Var}_{X \sim \mathcal{U}(\mathcal{S}^{d-1})}\left[e^{\kappa\langle V , X\rangle}\right]$ Although we do not need an explicit expression for $\sigma$, we know it is finite. Additionally, let $g : x \mapsto \dfrac{1}{x}$ be the inverse function. The CLT ensures that:

$$\sqrt{n}\Big[D_n - \frac{1}{\mathcal{A}(\mathcal{S}^{d-1})C_d(\kappa)}\Big] \xrightarrow{D} \mathcal{N}(0, \sigma^2), \tag{16}$$

where $\xrightarrow{D}$ denotes convergence in distribution (Jacod and Protter, 2004). Moreover, since $g$ is a differentiable function on $\mathbb{R}_+^*$, we use the *Delta method* (Oehlert, 1992) to infer that:

$$\sqrt{n}[g(D_n) - g(\frac{1}{\mathcal{A}(\mathcal{S}^{d-1})C_d(\kappa)})] \xrightarrow{D} \mathcal{N}(0, \sigma^2[g'(\frac{1}{\mathcal{A}(\mathcal{S}^{d-1})C_d(\kappa)})]^2). \tag{17}$$

Replacing $g$ and $g'$ by their respective values, we obtain:

$$\sqrt{n}\Big[\frac{1}{D_n} - C_d(\kappa)\mathcal{A}(\mathcal{S}^{d-1})\Big] \xrightarrow{D} \mathcal{N}(0, \sigma^2(\mathcal{A}(\mathcal{S}^{d-1})C_d(\kappa))^4). \tag{18}$$

Furthermore, recall that if a sequence $Z_1, Z_2, ...$ of random variables converges in distribution to a random variable $Z$, then for all bounded continuous function $\phi$, $\lim_{n \to +\infty} \mathbb{E}\left[\phi(Z_n)\right] = \mathbb{E}\left[\phi(Z)\right]$ (Jacod and Protter, 2004). Since for every $n$ the random variable $Z_n = \sqrt{n}[\frac{1}{D_n} - C_d(\kappa)\mathcal{A}(\mathcal{S}^{d-1})]$ has bounded values, we can simply chose the identity function for $\phi$ to conclude that :

$$\lim_{n \to +\infty} \mathbb{E}_{\mathcal{X}_n \sim \mathcal{U}(\mathcal{S}^{d-1})}\left[\sqrt{n}[\frac{1}{D_n} - C_d(\kappa)\mathcal{A}(\mathcal{S}^{d-1})]\right] = 0, \tag{19}$$

which is equivalent to:

$$\mathbb{E}_{\mathcal{X}_n \sim \mathcal{U}(\mathcal{S}^{d-1})}\left[\frac{1}{D_n}\right] = C_d(\kappa)\mathcal{A}(\mathcal{S}^{d-1}) + o(\frac{1}{\sqrt{n}}). \tag{20}$$

Finally, by multiplying Equation (20) by $\frac{e^{\kappa \langle V, A \rangle}}{n}$, we obtain Equation (11), concluding the proof. $\square$

## B  ASYMPTOTIC BEHAVIOR OF vMF EXPLORATION IN $d = 2$ DIMENSIONS (PROOF OF PROPOSITION 4.3, PART 1)

We now prove Proposition 4.3 when $d = 2$. In 2 dimensions, the vMF distribution takes the special form of the von Mises (vM) distribution (Mardia and Jupp, 2009) which, instead of describing the distribution of the dot product between $V$ and $\tilde{V}$, describes the distribution of their angle $\theta$. The PDF of a von Mises distribution is defined as follows:

$$\forall \theta \in [-\pi, \pi], f_{\mathrm{vM}}(\theta \mid \kappa) = \frac{e^{\kappa \cos(\theta)}}{2\pi I_0(\kappa)}. \tag{21}$$

Let us define $\theta_0$ as the angle between $V$ and $A$. In this section, we prove that:

$$P_{\mathrm{vMF\text{-}exp}}(A \mid n, d = 2, \kappa) = \frac{e^{\kappa \cos(\theta_0)}}{n I_0(\kappa)} + \mathcal{O}(\frac{1}{n^2}). \tag{22}$$

*Proof.* By definition,

$$P_{\mathrm{vMF\text{-}exp}}(A \mid n, d = 2, \kappa) = \mathbb{E}_{\mathcal{X}_n \sim \mathcal{U}(\mathcal{S}^1)}\left[\mathbb{P}(\tilde{V} \in \mathcal{S}_{\mathrm{Voronoï}}(A \mid \mathcal{X}_{n+1}))\right], \tag{23}$$

where $\mathcal{S}_{\mathrm{Voronoï}}(X_i \mid \mathcal{X}_n) = \{\tilde{V} \in \mathcal{S}^{d-1}, \forall j \in \mathcal{I}_n, \langle \tilde{V}, X_i \rangle \geq \langle \tilde{V}, X_j \rangle\}$. Let us call $\mathcal{Y}_n = \{Y_i\}$ the result of the permutation of the indices of $\mathcal{X}_n$ such that the (signed) angles $\beta_i$ between $A$ and $Y_i$ are sorted in increasing order. Since the $\{X_i\}$ are i.i.d. and uniformly distributed on the circle, then the angles between $A$ and the $\{X_i\}$ are i.i.d. and uniformly distributed on $[0, 2\pi]$. Therefore, the set $\{\beta_i\}$ is the set of the order statistics of $n$ i.i.d. random variables uniformly distributed on $[0, 2\pi]$. Consequently, the set $\{\frac{\beta_i}{2\pi}\}$ is the set of the order statistics of $n$ i.i.d. random variables uniformly distributed on $[0, 1]$, which are known to follow Beta distributions (Gentle, 2009) defined as follows:

$$\forall 1 \leq i \leq n, \frac{\beta_i}{2\pi} \sim \mathrm{Beta}(i, n + 1 - i). \tag{24}$$

As a consequence, we have:

$$\mathbb{E}\left[\beta_1\right] = \frac{2\pi}{n + 1}, \tag{25}$$

$$\mathbb{E}\left[\beta_n\right] = \frac{2\pi n}{n + 1}, \tag{26}$$

$$\mathrm{Var}\left[\beta_1\right] = \mathrm{Var}\left[\beta_n\right] = \frac{4\pi^2 n}{(n + 1)^2(n + 2)}. \tag{27}$$

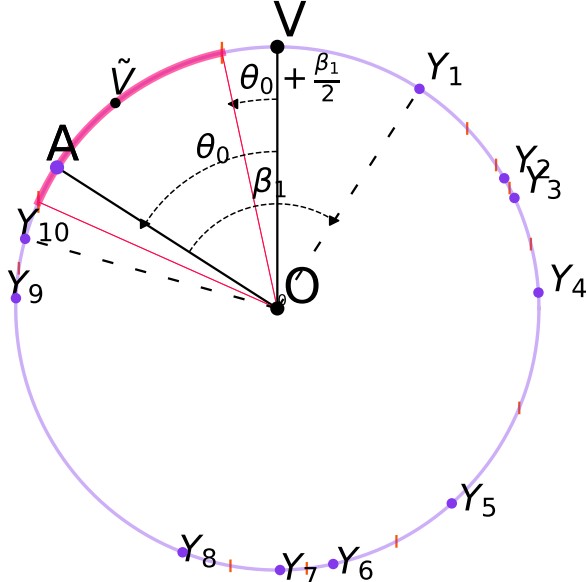

Figure 4: For $d = 2$: vMF-exp explores the action $A$ when $\tilde{V}$ lies in its Voronoï cell, shown in red.

Moreover, for given values of $Y_i$, we can see from Figure 4 that, in 2 dimensions, Voronoï cells are arcs of the circle and are delimited by perpendicular bisectors of two neighboring points. Specifically, the Voronoï cell of $A$ is delimited by the perpendicular bisector of A and $Y_1$ on one side, and the perpendicular bisector of A and $Y_n$ on the other side. By denoting $\theta$ the (signed) angle between $V$ and $\tilde{V}$, we have:

$$\mathbb{P}\Big(\tilde{V} \in \mathcal{S}_{\text{Voronoï}}(A \mid \mathcal{X}_{n+1})\Big) = \mathbb{P}\Big(\theta \in [\theta_0 + \frac{\beta_n - 2\pi}{2}, \theta_0 + \frac{\beta_1}{2}] \mid \theta \sim \text{vM}(0, \kappa), \beta_1, \beta_n\Big)$$
$$= \int_{\theta=\theta_0 + \frac{\beta_n - 2\pi}{2}}^{\theta_0 + \frac{\beta_1}{2}} f_{\text{vM}}(\theta \mid \kappa) \, d\theta. \tag{28}$$

Therefore:

$$P_{\text{vMF-exp}}(A \mid n, d = 2, \kappa) = \mathbb{E}_{\beta_1, \beta_n} \left[ \int_{\theta=\theta_0 + \frac{\beta_n - 2\pi}{2}}^{\theta_0 + \frac{\beta_1}{2}} f_{\text{vM}}(\theta \mid \kappa) \, d\theta \right]. \tag{29}$$

To get an asymptotic expression of the probability that $\theta$ lies between the considered bounds, we can first notice that as $n$ grows, $\beta_1$ will approach 0 and $\beta_n$ will approach $2\pi$. This means that the integral we need to compute will have very narrow bounds centered on $\theta_0$, and so we can leverage the Taylor series expansion (Abramowitz and Stegun, 1948) of $f_{\text{vM}}$ around $\theta_0$ and obtain:

$$f_{\text{vM}}(\theta \mid \kappa) = f_{\text{vM}}(\theta_0 \mid \kappa) + R_0(\theta), \tag{30}$$

where $R_0(\theta) = \sum_{i=1}^{\infty} \frac{f_{\text{vM}}^{(i)}(\theta_0 \mid \kappa)}{i!}(\theta - \theta_0)^i$ is the zero order remainder term of the Taylor series expansion of $f_{\text{vM}}$ near $\theta_0$.

We can now estimate the portion of the integral of Equation (29) corresponding to each term of the expansion separately, and show that when $n$ becomes large:

- the zero-order term gives a probability of selecting $A$ that is the same as the asymptotic behavior of B-exp: $\mathbb{E}_{\beta_1, \beta_n} \left[ \int_{\theta=\theta_0 + \frac{\beta_n - 2\pi}{2}}^{\theta_0 + \frac{\beta_1}{2}} f_{\text{vM}}(\theta_0 \mid \kappa) \, d\theta \right] = \frac{e^{\kappa \cos(\theta_0)}}{n I_0(\kappa)} + \mathcal{O}(\frac{1}{n^2})$.

- the expectation of the remainder term is bounded by a $\frac{1}{n^2}$ term: $\mathbb{E}_{\beta_1, \beta_n} \left[ \int_{\theta=\theta_0 + \frac{\beta_n - 2\pi}{2}}^{\theta_0 + \frac{\beta_1}{2}} R_0(\theta) \, d\theta \right] = \mathcal{O}(\frac{1}{n^2})$.

## B.1 ZERO-ORDER ESTIMATE

Let us study the zero-order approximation of $f_{\text{vM}}(\theta \mid \kappa)$ near $\theta_0$:

$$
\begin{aligned}
\mathbb{E}_{\beta_1,\beta_n}\left[\int_{\theta=\theta_0+\frac{\beta_n-2\pi}{2}}^{\theta_0+\frac{\beta_1}{2}} f_{\text{vM}}(\theta_0 \mid \kappa)\,\mathrm{d}\theta\right] &= \mathbb{E}_{\beta_1,\beta_n}\left[f_{\text{vM}}(\theta_0 \mid \kappa)(\theta_0 + \frac{\beta_1}{2} - (\theta_0 + \frac{\beta_n-2\pi}{2}))\right] \\
&= \mathbb{E}_{\beta_1,\beta_n}\left[f_{\text{vM}}(\theta_0 \mid \kappa)(\pi - \frac{\beta_n-\beta_1}{2})\right] \\
&= \pi f_{\text{vM}}(\theta_0 \mid \kappa)\,\mathbb{E}_{\beta_1,\beta_n}\left[1 - \frac{\beta_n-\beta_1}{2\pi}\right] \\
&= \frac{e^{\kappa\cos(\theta)}}{2I_0(\kappa)}(1 - \mathbb{E}_{\beta_1,\beta_n}\left[\frac{\beta_n}{2\pi}\right] + \mathbb{E}_{\beta_1,\beta_n}\left[\frac{\beta_1}{2\pi}\right]) \\
&= \frac{e^{\kappa\cos(\theta_0)}}{2I_0(\kappa)}\frac{n+1-n+1}{n+1} \\
&= \frac{e^{\kappa\cos(\theta_0)}}{2I_0(\kappa)}\frac{2}{n+1} \\
&= \frac{e^{\kappa\cos(\theta_0)}}{(n+1)I_0(\kappa)} \\
&= \frac{e^{\kappa\cos(\theta_0)}}{nI_0(\kappa)} - \frac{e^{\kappa\cos(\theta_0)}}{n(n+1)I_0(\kappa)} \\
&= \frac{e^{\kappa\cos(\theta_0)}}{nI_0(\kappa)} + \mathcal{O}(\frac{1}{n^2}).
\end{aligned}
\tag{31}
$$

This proves that, asymptotically, the contribution of the zero-order term of $f_{\text{vM}}$ to the probability of selecting $A$ is equal to the probability of selecting $A$ using B-exp with the same $\kappa$ value.

To understand how fast vMF-exp reaches its asymptotic behavior, we now need to study $R_0(\theta)$, the remainder of the Taylor series expansion of $f_{\text{VM}}$ around $\theta_0$.

## B.2 BOUNDING OF THE REMAINDER TERM

We start by computing the first derivative of $f_{\text{vM}}$:

$$
\forall \theta \in [0, 2\pi], |f_{\text{vM}}'(\theta \mid \kappa)| = \frac{|\sin(\theta)|\kappa e^{\kappa\cos(\theta)}}{I_0(\kappa)},
\tag{32}
$$

which is bounded[3] on $[0, 2\pi]$ by $M = \frac{\kappa e^{\kappa}}{I_0(\kappa)}$. According to the Taylor-Lagrange inequality (Abramowitz and Stegun, 1948), this in turn bounds the remainder term as follows:

$$
\forall \theta \in [0, 2\pi], |R_0(\theta)| \le M|\theta - \theta_0|.
\tag{33}
$$

---

[3]We note that a tighter bound could be found by studying the second derivative, but will not be necessary for the purpose of this proof.

In particular, this inequality holds for every $\theta \in [\theta_0 + \frac{\beta_n - 2\pi}{2}, \theta_0 + \frac{\beta_1}{2}]$, and so:

$$
\begin{aligned}
\int_{\theta=\theta_0+\frac{\beta_n-2\pi}{2}}^{\theta_0+\frac{\beta_1}{2}} |R_0(\theta)|\, \mathrm{d}\theta &\leq \int_{\theta=\theta_0+\frac{\beta_n-2\pi}{2}}^{\theta_0+\frac{\beta_1}{2}} M|\theta - \theta_0|\, \mathrm{d}\theta \\
&= \int_{\theta=\theta_0}^{\theta_0+\frac{\beta_1}{2}} M(\theta - \theta_0)\, \mathrm{d}\theta + \int_{\theta=\theta_0+\frac{\beta_n-2\pi}{2}}^{\theta_0} M(\theta_0 - \theta)\, \mathrm{d}\theta \\
&= \int_{\theta=0}^{\frac{\beta_1}{2}} M\theta\, \mathrm{d}\theta - \int_{\theta=\frac{\beta_n-2\pi}{2}}^{0} M\theta\, \mathrm{d}\theta \\
&= M\frac{\beta_1^2 + (\beta_n - 2\pi)^2}{8}.
\end{aligned}
\tag{34}
$$

The above inequality holds when considering the expected values over uniformly distributed $X_i$:

$$
\begin{aligned}
\mathbb{E}_{\beta_1,\beta_n}\left[\int_{\theta=\theta_0+\frac{\beta_n-2\pi}{2}}^{\theta_0+\frac{\beta_1}{2}} |R_0(\theta)|\, \mathrm{d}\theta\right] &\leq M\frac{\mathbb{E}_{\beta_1,\beta_n}\left[\beta_1^2\right] + \mathbb{E}_{\beta_1,\beta_n}\left[(\beta_n - 2\pi)^2\right]}{8} \\
&= M\frac{\mathrm{Var}_{\beta_1,\beta_n}\left[\beta_1\right] + (\mathbb{E}_{\beta_1,\beta_n}\left[\beta_1\right])^2 + \mathrm{Var}_{\beta_1,\beta_n}\left[(\beta_n - 2\pi)\right] + (\mathbb{E}_{\beta_1,\beta_n}\left[\beta_n - 2\pi\right])^2}{8} \\
&= \frac{M}{8}\left(\frac{2 \times 4\pi^2 n}{(n+1)^2(n+2)} + \frac{2 \times 4\pi^2}{(n+1)^2}\right) \\
&= \frac{M\pi^2}{(n+1)(n+2)} \\
&= \mathcal{O}(\frac{1}{n^2}).
\end{aligned}
\tag{35}
$$

Since $\left|\mathbb{E}_{\beta_1,\beta_n}\left[\int_{\theta=\theta_0+\frac{\beta_n-2\pi}{2}}^{\theta_0+\frac{\beta_1}{2}} R_0(\theta)\, \mathrm{d}\theta\right]\right| \leq \mathbb{E}_{\beta_1,\beta_n}\left[\int_{\theta=\theta_0+\frac{\beta_n-2\pi}{2}}^{\theta_0+\frac{\beta_1}{2}} |R_0(\theta)|\, \mathrm{d}\theta\right]$, we have shown:

$$
\mathbb{E}_{\beta_1,\beta_n}\left[\int_{\theta=\theta_0+\frac{\beta_n-2\pi}{2}}^{\theta_0+\frac{\beta_1}{2}} R_0(\theta)\, \mathrm{d}\theta\right] = \mathcal{O}(\frac{1}{n^2}).
\tag{36}
$$

In summary, when combining the asymptotic behavior of the zero-order term and the remainder term, we conclude that when $d = 2$ we have:

$$
P_{\text{vMF-exp}}(A \mid n, d = 2, \kappa) = \frac{\mathrm{e}^{\kappa \cos(\theta_0)}}{nI_0(\kappa)} + \mathcal{O}(\frac{1}{n^2}).
\tag{37}
$$

This proves Proposition 4.3 when $d = 2$. Note that, comparing the asymptotic expressions for $P_{\text{B-exp}}(A \mid n, d = 2, \kappa)$ and $P_{\text{vMF-exp}}(A \mid n, d = 2, \kappa)$, also gives us a proof for Proposition 4.1 when $d = 2$. $\qquad\square$

# C  ASYMPTOTIC BEHAVIOR OF vMF EXPLORATION IN $d > 2$ DIMENSIONS (PROOFS OF PROPOSITION 4.3, PART 2, AND OF PROPOSITION 4.4)

We now prove Proposition 4.3 when $d > 2$, starting with a series of intermediary lemmas. We subsequently justify the approximate expression of Proposition 4.4.

## C.1  INTERMEDIARY LEMMAS

We introduce a series of lemmas regarding the properties of the Voronoï cell of $A$ when $\mathcal{X}_n \sim \mathcal{U}^{d-1}$. We recall that, for a given set of embedding vectors $\mathcal{X}_n$, we use the notation $\mathcal{X}_{n+1} = \mathcal{X}_n \cup \{A\}$.

**Lemma C.1.** *Let $d \in \mathbb{N}, d \geq 2$, $A \in \mathcal{S}^{d-1}$ and $n \in \mathbb{N}^*$. As before, let $\mathcal{A}(\mathcal{S}^{d-1})$ denote the surface area of $\mathcal{S}^{d-1}$. Then:*

$$
\mathbb{E}_{\mathcal{X}_n \sim \mathcal{U}(\mathcal{S}^{d-1})}\left[\mathcal{A}(\mathcal{S}_{Voronoï}(A \mid \mathcal{X}_{n+1}))\right] = \frac{\mathcal{A}(\mathcal{S}^{d-1})}{n+1}.
\tag{38}
$$

*Proof.* To compute this expectation, one can notice that:

$$\mathbb{E}_{\mathcal{X}_n \sim \mathcal{U}(\mathcal{S}^{d-1})}\Big[\mathcal{A}(\mathcal{S}_{\text{Voronoï}}(A \mid \mathcal{X}_{n+1}))\Big] = \mathbb{E}_{\mathcal{X}_{n+1} \sim \mathcal{U}(\mathcal{S}^{d-1})}\Big[\mathcal{A}(\mathcal{S}_{\text{Voronoï}}(X_{n+1} \mid \mathcal{X}_{n+1})) \mid X_{n+1} = A\Big]. \tag{39}$$

Indeed, considering that $A$ is known is equivalent to considering $A$ as a random vector $X_{n+1} \sim \mathcal{U}(\mathcal{S}^{d-1})$ with the constraint $X_{n+1} = A$. We will now show that the right part of Equation (39) is actually independent of the value of $A$.

Consider any point $A' \in \mathcal{S}^{d-1}$. One can always define a (not necessarily unique) rotation $R_{A,A'}$ such that $R_{A,A'}(A) = A'$. Since rotations preserve inner products, they also preserve areas of Voronoï cells, which means that for a given set of vectors $\mathcal{X}_{n+1}$, we have:

$$\mathcal{A}\Big(\mathcal{S}_{\text{Voronoï}}(X_{n+1} \mid \mathcal{X}_{n+1})\Big) = \mathcal{A}\Big(\mathcal{S}_{\text{Voronoï}}(R_{A,A'}(X_{n+1}) \mid R_{A,A'}(\mathcal{X}_{n+1}))\Big). \tag{40}$$

Moreover, the image of the rotation of a random vector uniformly distributed on the hypersphere is also uniformly distributed, which means that:

$$\mathcal{X}_{n+1} \sim \mathcal{U}(\mathcal{S}^{d-1}) \Leftrightarrow R_{A,A'}(\mathcal{X}_{n+1}) \sim \mathcal{U}(\mathcal{S}^{d-1}). \tag{41}$$

Therefore:

$$\begin{aligned}
&\mathbb{E}_{\mathcal{X}_{n+1} \sim \mathcal{U}(\mathcal{S}^{d-1})}\Big[\mathcal{A}(\mathcal{S}_{\text{Voronoï}}(X_{n+1} \mid \mathcal{X}_{n+1})) \mid X_{n+1} = A\Big] \\
&= \mathbb{E}_{\mathcal{X}_{n+1} \sim \mathcal{U}(\mathcal{S}^{d-1})}\Big[\mathcal{A}(\mathcal{S}_{\text{Voronoï}}(R_{A,A'}(X_{n+1}) \mid R_{A,A'}(\mathcal{X}_{n+1}))) \mid X_{n+1} = A\Big] \\
&= \mathbb{E}_{R_{A,A'}(\mathcal{X}_{n+1}) \sim \mathcal{U}(\mathcal{S}^{d-1})}\Big[\mathcal{A}(\mathcal{S}_{\text{Voronoï}}(R_{A,A'}(X_{n+1}) \mid R_{A,A'}(\mathcal{X}_{n+1}))) \mid R_{A,A'}(X_{n+1}) = A'\Big] \\
&= \mathbb{E}_{R_{A,A'}(\mathcal{X}_n) \sim \mathcal{U}(\mathcal{S}^{d-1})}\Big[\mathcal{A}(\mathcal{S}_{\text{Voronoï}}(A' \mid R_{A,A'}(\mathcal{X}_n)))\Big] \\
&= \mathbb{E}_{\mathcal{X}_n \sim \mathcal{U}(\mathcal{S}^{d-1})}\Big[\mathcal{A}(\mathcal{S}_{\text{Voronoï}}(A' \mid \mathcal{X}_n))\Big].
\end{aligned} \tag{42}$$

This result proves that $\mathbb{E}_{\mathcal{X}_n \sim \mathcal{U}(\mathcal{S}^{d-1})}[\mathcal{A}(\mathcal{S}_{\text{Voronoï}}(A \mid \mathcal{X}_{n+1}))]$ is independent of $A$. Then, we use this information along with Equation (39) to obtain:

$$\mathbb{E}_{\mathcal{X}_n \sim \mathcal{U}(\mathcal{S}^{d-1})}\Big[\mathcal{A}(\mathcal{S}_{\text{Voronoï}}(A \mid \mathcal{X}_{n+1}))\Big] = \mathbb{E}_{\mathcal{X}_{n+1} \sim \mathcal{U}(\mathcal{S}^{d-1})}\Big[\mathcal{A}(\mathcal{S}_{\text{Voronoï}}(X_{n+1} \mid \mathcal{X}_{n+1}))\Big]. \tag{43}$$

Since $\sum_{i=1}^{n+1} \mathcal{A}(\mathcal{S}_{\text{Voronoï}}(X_i \mid \mathcal{X}_{n+1})) = \mathcal{A}(\mathcal{S}^{d-1})$ (Du et al., 1999; 2010) and the $X_i$ are i.i.d., we derive:

$$\mathbb{E}_{\mathcal{X}_{n+1} \sim \mathcal{U}(\mathcal{S}^{d-1})}\Big[\mathcal{A}(\mathcal{S}_{\text{Voronoï}}(X_{n+1} \mid \mathcal{X}_{n+1}))\Big] = \frac{\mathcal{A}(\mathcal{S}^{d-1})}{n+1}. \tag{44}$$

Combining Equations (39) with Equation (44) leads to Equation (38), concluding the proof. $\qquad \square$

**Lemma C.2.** *Let $d \in \mathbb{N}, d \geq 2$, $A \in \mathcal{S}^{d-1}$ and $n \in \mathbb{N}^*$. Then:*

$$\exists \lambda \in \mathbb{R}, \mathbb{E}_{\mathcal{X}_n \sim \mathcal{U}(\mathcal{S}^{d-1})}\Big[\int_{\tilde{V} \in \mathcal{S}_{Voronoï}(A \mid \mathcal{X}_{n+1})} \tilde{V}\, \mathrm{d}\tilde{V}\Big] = \lambda A. \tag{45}$$

*Proof.* We want to prove that the average normal vector of the Voronoï cell of $A$ and $A$ are collinear, as illustrated in Figure 5. To do so, we will show that this average normal vector is invariant to any rotation around $A$. For every $\theta \in [0, 2\pi]$, we define $R_{A,\theta}$ as the rotation around $A$ of the angle $\theta$. As discussed in the proof of Lemma C.1, $\mathcal{X}_n \sim \mathcal{U}(\mathcal{S}^{d-1}) \Leftrightarrow R_{A,\theta}(\mathcal{X}_n) \sim \mathcal{U}(\mathcal{S}^{d-1})$. Moreover, $R_{A,\theta}(A) = A$. Let us denote:

$$N(A \mid n) = \mathbb{E}_{\mathcal{X}_n \sim \mathcal{U}(\mathcal{S}^{d-1})}\Big[\int_{\tilde{V} \in \mathcal{S}_{\text{Voronoï}}(A \mid \mathcal{X}_{n+1})} \tilde{V}\, \mathrm{d}\tilde{V}\Big], \tag{46}$$

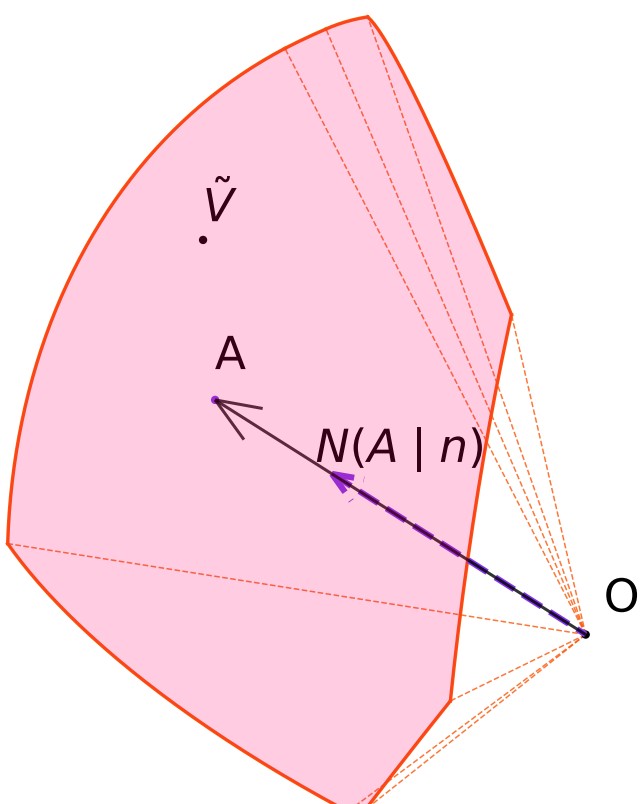

Figure 5: The Voronoï cell of $A$, $\mathcal{S}_{\mathrm{Voronoï}}(A \mid \mathcal{X}_{n+1})$, along with the average normal vector of the cell $N(A \mid n)$. On expectation, $(A \mid n)$ and $A$ are collinear.

the expected normal vector of the Voronoï cell of $A$. Its image by the rotation $R_{A,\theta}$ verifies:

$$
\begin{aligned}
R_{A,\theta}(N(A \mid n)) &= R_{A,\theta}\Big( \mathbb{E}_{\ \mathcal{X}_n \sim \mathcal{U}(\mathcal{S}^{d-1})} \Big[ \int_{\tilde{V} \in \mathcal{S}_{\mathrm{Voronoï}}(A \mid \mathcal{X}_{n+1})} \tilde{V} \, \mathrm{d}\tilde{V} \Big] \Big) \\
&= \mathbb{E}_{\ \mathcal{X}_n \sim \mathcal{U}(\mathcal{S}^{d-1})} \Big[ \int_{\tilde{V} \in \mathcal{S}_{\mathrm{Voronoï}}(R_{A,\theta}(A) \mid R_{A,\theta}(\mathcal{X}_{n+1}))} \tilde{V} \, \mathrm{d}\tilde{V} \Big] \\
&= \mathbb{E}_{\ R_{A,\theta}(\mathcal{X}_n) \sim \mathcal{U}(\mathcal{S}^{d-1})} \Big[ \int_{\tilde{V} \in \mathcal{S}_{\mathrm{Voronoï}}(A \mid R_{A,\theta}(\mathcal{X}_{n+1}))} \tilde{V} \, \mathrm{d}\tilde{V} \Big] \\
&= N(A \mid n).
\end{aligned}
\tag{47}
$$

This proves that $N(A \mid n)$ and $A$ are collinear. $\qquad\square$

**Lemma C.3.** *With the same hypotheses as Lemma C.2:*

$$
\lambda = \frac{\mathcal{A}(\mathcal{S}^{d-1})}{n+1} \mathbb{E}_{\ \mathcal{X}_n \sim \mathcal{U}(\mathcal{S}^{d-1}), \tilde{V} \sim \mathcal{U}(\mathcal{S}^{d-1})} \Big[ \max_i \langle \tilde{V} \ , \ X_i \rangle \Big].
\tag{48}
$$

*Proof.* $\lambda$ is defined as follows:

$$\lambda A = \mathbb{E}_{\mathcal{X}_n \sim \mathcal{U}(\mathcal{S}^{d-1})}\Big[\int_{\tilde{V} \in \mathcal{S}_{\text{Voronoï}}(A|\mathcal{X}_{n+1})} \tilde{V}\, \mathrm{d}\tilde{V}\Big]$$

$$\implies \langle \lambda A\, ,\, A\rangle = \langle \mathbb{E}_{\mathcal{X}_n \sim \mathcal{U}(\mathcal{S}^{d-1})}\Big[\int_{\tilde{V} \in \mathcal{S}_{\text{Voronoï}}(A|\mathcal{X}_{n+1})} \tilde{V}\, \mathrm{d}\tilde{V}\Big]\, ,\, A\rangle$$

$$\Leftrightarrow \qquad \lambda = \mathbb{E}_{\mathcal{X}_n \sim \mathcal{U}(\mathcal{S}^{d-1})}\Big[\int_{\tilde{V} \in \mathcal{S}_{\text{Voronoï}}(A|\mathcal{X}_{n+1})} \langle \tilde{V}\, ,\, A\rangle\, \mathrm{d}\tilde{V}\Big] \qquad (49)$$

$$\Leftrightarrow \qquad \lambda = \mathbb{E}_{\mathcal{X}_{n+1} \sim \mathcal{U}(\mathcal{S}^{d-1})}\Big[\int_{\tilde{V} \in \mathcal{S}_{\text{Voronoï}}(X_{n+1}|\mathcal{X}_{n+1})} \langle \tilde{V}\, ,\, X_{n+1}\rangle\, \mathrm{d}\tilde{V} \mid X_{n+1} = A\Big]$$

$$\Leftrightarrow \qquad \lambda = \mathbb{E}_{\mathcal{X}_{n+1} \sim \mathcal{U}(\mathcal{S}^{d-1})}\Big[\int_{\tilde{V} \in \mathcal{S}_{\text{Voronoï}}(X_{n+1}|\mathcal{X}_{n+1})} \max_i \langle \tilde{V}\, ,\, X_i\rangle\, \mathrm{d}\tilde{V} \mid X_{n+1} = A\Big].$$

Moreover, as done in the proof of Lemma C.1, we can leverage the invariance by any rotation of the above expression to infer that the conditional expectation is actually independent of $A$:

$$\lambda = \mathbb{E}_{\mathcal{X}_{n+1} \sim \mathcal{U}(\mathcal{S}^{d-1})}\Big[\int_{\tilde{V} \in \mathcal{S}_{\text{Voronoï}}(X_{n+1}|\mathcal{X}_{n+1})} \max_i \langle \tilde{V}\, ,\, X_i\rangle\, \mathrm{d}\tilde{V}\Big]. \qquad (50)$$

Since, in the above equation, $X_{n+1}$ has the same distribution as every element of $\mathcal{X}_{n+1}$, a similar expression for $\lambda$ can be found using each $\mathcal{X}_{n+1}$ element. By summing them together, we obtain:

$$(n+1)\lambda = \sum_{j=1}^{n+1} \mathbb{E}_{\mathcal{X}_{n+1} \sim \mathcal{U}(\mathcal{S}^{d-1})}\Big[\int_{\tilde{V} \in \mathcal{S}_{\text{Voronoï}}(X_j|\mathcal{X}_{n+1})} \max_i \langle \tilde{V}\, ,\, X_i\rangle\, \mathrm{d}\tilde{V}\Big]$$

$$= \mathbb{E}_{\mathcal{X}_{n+1} \sim \mathcal{U}(\mathcal{S}^{d-1})}\Big[\sum_{j=1}^{n+1}\int_{\tilde{V} \in \mathcal{S}_{\text{Voronoï}}(X_j|\mathcal{X}_{n+1})} \max_i \langle \tilde{V}\, ,\, X_i\rangle\, \mathrm{d}\tilde{V}\Big]$$

$$= \mathbb{E}_{\mathcal{X}_{n+1} \sim \mathcal{U}(\mathcal{S}^{d-1})}\Big[\int_{\tilde{V} \in \mathcal{S}^{d-1}} \max_i \langle \tilde{V}\, ,\, X_i\rangle\, \mathrm{d}\tilde{V}\Big] \qquad (51)$$

$$= \mathbb{E}_{\mathcal{X}_{n+1} \sim \mathcal{U}(\mathcal{S}^{d-1})}\Big[\int_{\tilde{V} \in \mathcal{S}^{d-1}} \frac{\mathcal{A}(\mathcal{S}^{d-1})\max_i \langle \tilde{V}\, ,\, X_i\rangle}{\mathcal{A}(\mathcal{S}^{d-1})}\, \mathrm{d}\tilde{V}\Big]$$

$$= \mathcal{A}(\mathcal{S}^{d-1})\,\mathbb{E}_{\mathcal{X}_{n+1} \sim \mathcal{U}(\mathcal{S}^{d-1})}\Big[\mathbb{E}_{\tilde{V} \sim \mathcal{U}(\mathcal{S}^{d-1})}[\max_i \langle \tilde{V}\, ,\, X_i\rangle]\Big],$$

which proves the lemma. $\qquad\qquad\square$

The last two lemmas are useful to describe the distribution of $\max_i \langle \tilde{V}\, ,\, X_i\rangle$ when $\tilde{V}$ is fixed, $\mathcal{X}_{n+1} \sim \mathcal{U}(\mathcal{S}^{d-1})$, and $n$ is large.

**Lemma C.4.** *Let* $B : (z_1, z_2) \mapsto \int_0^1 t^{z_1-1}(1-t)^{z_2-1}\, \mathrm{d}t$ *denote the Beta function. Let* $d \geq 3$, $\tilde{V} \in \mathcal{S}^{d-1}$ *and* $X$ *be a random vector with* $X \sim \mathcal{U}(\mathcal{S}^{d-1})$. *Let* $F_{radial}$ *be the cumulative distribution function (CDF) of* $\langle \tilde{V}\, ,\, X\rangle$. *The Taylor series expansion of* $F_{radial}$ *near 1 is:*

$$F_{radial}(t) = 1 - \frac{2^{\frac{d-1}{2}}}{(d-1)B(\frac{1}{2}, \frac{d-1}{2})}(1-t)^{\frac{d-1}{2}} + o((1-t)^{\frac{d-1}{2}}). \qquad (52)$$

*Proof.* The distribution of $\langle \tilde{V} , X \rangle$ has been studied in directional statistics (Mardia and Jupp, 2009). Its PDF is known to be:

$$
\begin{aligned}
f_{\text{radial}}(t) &= \frac{(1-t^2)^{\frac{d-1}{2}-1}}{B(\frac{1}{2}, \frac{d-1}{2})} \\
&= \frac{(1-t)^{\frac{d-1}{2}-1}(1+t)^{\frac{d-1}{2}-1}}{B(\frac{1}{2}, \frac{d-1}{2})} \\
&= \frac{(1-t)^{\frac{d-1}{2}-1}(2-(1-t))^{\frac{d-1}{2}-1}}{B(\frac{1}{2}, \frac{d-1}{2})} \\
&= \frac{2^{\frac{d-1}{2}-1}(1-t)^{\frac{d-1}{2}-1}(1-\frac{(1-t)}{2})^{\frac{d-1}{2}-1}}{B(\frac{1}{2}, \frac{d-1}{2})} \\
&= \frac{2^{\frac{d-1}{2}-1}(1-t)^{\frac{d-1}{2}-1}}{B(\frac{1}{2}, \frac{d-1}{2})} \left( \sum_{i=0}^{=\infty} \binom{\frac{d-1}{2}-1}{i} \left( \frac{1-t}{2} \right)^i \right).
\end{aligned}
\tag{53}
$$

The last line above was obtained using Newton's generalized binomial theorem for real exponent (Coolidge, 1949). It involves the term $\binom{\frac{d-1}{2}-1}{i} = \frac{(\frac{d-1}{2}-1)_i}{i!}$ with $(\cdot)_i$ the Pochhammer symbol used to designate a falling factorial (Abramowitz and Stegun, 1948). We have obtained an expression of $f_{\text{radial}}$ involving an infinite weighted sum of powers of $(1-t)$ with exponents greater or equal to 0 since $d \geq 3$. Therefore, by uniqueness of the Taylor polynomial, we derive that the Taylor series expansion of $f_{\text{radial}}$ near 1 is:

$$
f_{\text{radial}}(t) = \frac{2^{\frac{d-1}{2}-1}(1-t)^{\frac{d-1}{2}-1}}{B(\frac{1}{2}, \frac{d-1}{2})} + o((1-t)^{\frac{d-1}{2}-1}).
\tag{54}
$$

Since by definition $F_{\text{radial}}$ is the primitive of $f_{\text{radial}}$ on $[-1,1]$ and that $F_{\text{radial}}(1) = 1$, we can integrate the above equation to get:

$$
\begin{aligned}
F_{\text{radial}}(t) &= 1 - \frac{2}{d-1} \frac{2^{\frac{d-1}{2}-1}(1-t)^{\frac{d-1}{2}}}{B(\frac{1}{2}, \frac{d-1}{2})} + o((1-t)^{\frac{d-1}{2}}) \\
&= 1 - \frac{2^{\frac{d-1}{2}}(1-t)^{\frac{d-1}{2}}}{(d-1)B(\frac{1}{2}, \frac{d-1}{2})} + o((1-t)^{\frac{d-1}{2}}).
\end{aligned}
\tag{55}
$$

Since this is exactly the Equation (52), this completes the proof. $\qquad\square$

**Lemma C.5.** *Let $d \geq 3$, $\tilde{V} \in \mathcal{S}^{d-1}$ and let $F_{radial}$ be defined as in Lemma C.4. For $n \in \mathbb{N}^*$, let $\mathcal{X}_n \sim \mathcal{U}(\mathcal{S}^{d-1})$ be a set of $n$ i.i.d. random vectors uniformly distributed on $\mathcal{S}^{d-1}$, and let $F_n$ be the CDF of $\max_i \langle \tilde{V} , X_i \rangle$. Then, for $u \in [-1, 1]$:*

$$
\lim_{n \to +\infty} F_n(a_n u + b_n) = e^{(-(1+\gamma u)^{\frac{-1}{\gamma}})},
\tag{56}
$$

*where $\gamma = -\frac{2}{d-1}$, $a_n = \frac{1}{2} \left( \frac{(d-1)B(\frac{1}{2}, \frac{d-1}{2})}{n} \right)^{\frac{2}{d-1}}$ with $B$ the Beta function, and $b_n = 1 - \frac{2}{d-1}a_n$.*

*Proof.* The proof relies on the Fisher–Tippett–Gnedenko theorem (Gnedenko, 1943) which states that if there exists a couple of sequences $a_n$ and $b_n$ such that the left term of Equation (56) converges, then its limit should be the CDF of a Generalized Extreme Value distribution (GEV) with shape parameter $\gamma$, which is the right term of Equation (56). Theorem 5 of Gnedenko (1943) provides a necessary and sufficient convergence condition for a random variable with maximal value $x_{\max}$ and CDF F, provided that $\gamma < 0$:

$$
\lim_{t \to 0^+} \frac{1 - F(x_{\max} - ut)}{1 - F(x_{\max} - t)} = u^{\left( \frac{-1}{\gamma} \right)} \text{ for all } u > 0.
\tag{57}
$$

Recall that Lemma C.4 gives us the Taylor expansion of $F_{\text{radial}}$ near $1 : F_{\text{radial}}(t) = 1 - K(1-t)^{\frac{d-1}{2}} + o((1-t)^{\frac{d-1}{2}})$ with $K = \frac{2^{\frac{d-1}{2}}}{(d-1)B(\frac{1}{2}, \frac{d-1}{2})}$. Knowing that $x_{\max} = 1$, we obtain that, $\forall u > 0$:

$$
\begin{aligned}
\lim_{t \to 0^+} \frac{1 - F_{\text{radial}}(1 - u\,t\,)}{1 - F_{\text{radial}}(1 - t)} &= \lim_{t \to 0^+} \frac{1 - (1 - K(ut)^{\frac{d-1}{2}}) + o((t)^{\frac{d-1}{2}})}{1 - (1 - K(t)^{\frac{d-1}{2}}) + o((t)^{\frac{d-1}{2}})} \\
&= \lim_{t \to 0^+} \frac{K(ut)^{\frac{d-1}{2}} + o((t)^{\frac{d-1}{2}})}{K(t)^{\frac{d-1}{2}} + o((t)^{\frac{d-1}{2}})} \\
&= u^{\left(\frac{d-1}{2}\right)},
\end{aligned}
\tag{58}
$$

which guarantees convergence and in the same time gives the value of $\gamma = -\frac{2}{d-1}$.

To find suitable sequences $a_n$ and $b_n$, we can use the fact that $F_n(t) = F_{\text{radial}}(t)^n$ and study the behavior of $\ln F_n(t)$ near $t = 1$:

$$
\begin{aligned}
\ln F_n(t) &= \ln\left(F_{\text{radial}}(t)^n\right) \\
&= n \ln\left(F_{\text{radial}}(t)\right) \\
&= n(\ln\left(1 - K(1-t)^{\frac{-1}{\gamma}} + o((1-t)^{\frac{-1}{\gamma}})\right)) \text{ as } t \to 1^- \\
&= -nK((1-t)^{\frac{-1}{\gamma}} + o((1-t)^{\frac{-1}{\gamma}})) \text{ as } t \to 1^-.
\end{aligned}
\tag{59}
$$

By defining $a_n = -\gamma(Kn)^\gamma$, $b_n = 1 - (Kn)^\gamma$ and doing the change of variable $u = \frac{t - b_n}{a_n}$, we see that:

$$
\begin{aligned}
t &= a_n u + b_n \\
&= 1 - (1 + \gamma u)(Kn)^\gamma.
\end{aligned}
\tag{60}
$$

Since for every $u$, $\lim_{n \to +\infty}(1 + \gamma u)(Kn)^\gamma = 0$ (recall that $\gamma < 0$), the term $o((1-x)^{\frac{-1}{\gamma}})$ as $x \to 1^-$ is equivalent to $o(\frac{1}{n})$ as $n \to +\infty$. this means that:

$$
\begin{aligned}
\ln\left(F_n(a_n u + b_n)\right) &= -nK\left(((1 + \gamma u)(Kn)^\gamma)^{\frac{-1}{\gamma}} + o\left((\frac{1}{n})\right)\right) \text{ as } n \to +\infty. \\
&= -(1 + \gamma u)^{\frac{-1}{\gamma}} + o(1) \text{ as } n \to +\infty.
\end{aligned}
\tag{61}
$$

We can now consider the exponential of the above expression to get our asymptotic maximum distribution:

$$
\lim_{n \to +\infty} F_n(a_n u + b_n) = e^{-(1 + \gamma u)^{\frac{-1}{\gamma}}},
\tag{62}
$$

which concludes the proof. $\qquad\square$

**Corollary C.5.1.** *With $\Gamma : z \mapsto \int_0^\infty t^{z-1} e^{-t}\,dt$ the Gamma function (Abramowitz and Stegun, 1948), we have:*

$$
\mathbb{E}_{\mathcal{X}_n \sim \mathcal{U}(\mathcal{S}^{d-1})}\left[\max_i \langle \tilde{V}\,,\,X_i \rangle\right] = 1 - \frac{\Gamma(\frac{d+1}{d-1})}{2}\left(\frac{(d-1)B(\frac{1}{2}, \frac{d-1}{2})}{n}\right)^{\frac{2}{d-1}} + o(\frac{1}{n^{\frac{2}{d-1}}}).
\tag{63}
$$

*Proof.* According to the Portmanteau theorem (Billingsley, 2013), Lemma C.5 is equivalent to:

$$
\frac{\max_i \langle \tilde{V}\,,\,X_i \rangle - b_n}{a_n} \xrightarrow{D} \text{GEV}(\gamma),
\tag{64}
$$

where $\text{GEV}(\gamma)$ is a generalized extreme value distribution with shape parameter $\gamma$ (Gnedenko, 1943). Recall that if a sequence $Z_1, Z_2, \ldots$ of random variables converges in distribution a random variable $Z$, then for all bounded continuous function $\phi$, $\lim_{n \to +\infty} \mathbb{E}\left[\phi(Z_n)\right] = \mathbb{E}\left[\phi(Z)\right]$. Since $\frac{\max_i \langle \tilde{V}\,,\,X_i \rangle - b_n}{a_n}$ is bounded for every $n$, we can consider the identity function for $\phi$ and obtain:

$$
\lim_{n \to +\infty} \mathbb{E}_{\mathcal{X}_n \sim \mathcal{U}(\mathcal{S}^{d-1})}\left[\frac{\max_i \langle \tilde{V}\,,\,X_i \rangle - b_n}{a_n}\right] = \mathbb{E}\left[\text{GEV}(\gamma)\right] = \frac{\Gamma(1 - \gamma) - 1}{\gamma}.
\tag{65}
$$

Replacing $\gamma$, $a_n$ and $b_n$ by their respective expressions, it implies that:

$$\lim_{n\to+\infty} \frac{\mathbb{E}_{\mathcal{X}_n\sim\mathcal{U}(\mathcal{S}^{d-1})}\left[\max_i\langle\tilde{V}\,,\,X_i\rangle\right]-1+(Kn)^{-\frac{2}{d-1}}}{(Kn)^{-\frac{2}{d-1}}}+\Gamma(\frac{d-1}{d-1})-1=0$$

$$\implies \quad \lim_{n\to+\infty} \frac{\mathbb{E}_{\mathcal{X}_n\sim\mathcal{U}(\mathcal{S}^{d-1})}\left[\max_i\langle\tilde{V}\,,\,X_i\rangle\right]-1+K^{-\frac{2}{d-1}}\Gamma(\frac{d-1}{d-1})}{n^{-\frac{2}{d-1}}}=0. \tag{66}$$

Since $K^{-\frac{2}{d-1}}=\frac{1}{2}\left(\frac{(d-1)\mathrm{B}(\frac{1}{2},\frac{d-1}{2})}{n}\right)^{\frac{2}{d-1}}$, this is equivalent to writing:

$$\mathbb{E}_{\mathcal{X}_n\sim\mathcal{U}(\mathcal{S}^{d-1})}\left[\max_i\langle\tilde{V}\,,\,X_i\rangle\right]-1+\frac{1}{2}\left(\frac{(d-1)\mathrm{B}(\frac{1}{2},\frac{d-1}{2})}{n}\right)^{\frac{2}{d-1}}\Gamma(\frac{d-1}{d-1})=o(\frac{1}{n^{\frac{2}{d-1}}})$$

$$\Leftrightarrow \mathbb{E}_{\mathcal{X}_n\sim\mathcal{U}(\mathcal{S}^{d-1})}\left[\max_i\langle\tilde{V}\,,\,X_i\rangle\right]=1-\Gamma(\frac{d-1}{d-1})\frac{1}{2}\left(\frac{(d-1)\mathrm{B}(\frac{1}{2},\frac{d-1}{2})}{n}\right)^{\frac{2}{d-1}}+o(\frac{1}{n^{\frac{2}{d-1}}}). \tag{67}$$

We have thus obtained Equation (63), concluding the proof of the corollary. $\qquad\square$

## C.2 PROOF OF PROPOSITION 4.3

We now return to Proposition 4.3. In this section we consider the case of vMF-exp when $d>2$ and $X_i$ embeddings are uniformly distributed on $\mathcal{S}^{d-1}$. Under those assumptions:

$$P_{\text{vMF-exp}}(a\mid n,d,V,\kappa)=\frac{f_{\text{vMF}}(A\mid V,\kappa)\mathcal{A}(\mathcal{S}^{d-1})}{n}+\mathcal{O}(\frac{1}{n^{1+\frac{2}{d-1}}}). \tag{68}$$

*Proof.* Similarly to the 2 dimensional case, the definition of $P_{\text{vMF-exp}}(a\mid n,d,V,\kappa)$ is:

$$P_{\text{vMF-exp}}(A\mid n,d,V,\kappa)=\mathbb{E}_{\mathcal{X}_n\sim\mathcal{U}(\mathcal{S}^{d-1})}\left[\mathbb{P}(\tilde{V}\in\mathcal{S}_{\text{Voronoï}}(A\mid\mathcal{X}_{n+1})\mid\tilde{V}\sim\text{vMF}(V,\kappa))\right], \tag{69}$$

which can be written using the PDF of the vMF distribution:

$$P_{\text{vMF-exp}}(A\mid n,d,V,\kappa)=\mathbb{E}_{\mathcal{X}_n\sim\mathcal{U}(\mathcal{S}^{d-1})}\left[\int_{\tilde{V}\in\mathcal{S}_{\text{Voronoï}}(A\mid\mathcal{X}_{n+1})}f_{\text{vMF}}(\tilde{V}\mid V,\kappa)\,\mathrm{d}\tilde{V}\right]. \tag{70}$$

As done in the 2D case, we study the Taylor expansion of $f_{\text{vMF}}$ near $A$:

$$\forall\tilde{V}\in\mathcal{S}_{\text{Voronoï}}(A\mid\mathcal{X}_{n+1}), f_{\text{vMF}}(\tilde{V}\mid\kappa,V)=C_d(\kappa)e^{\kappa\langle V,\tilde{V}\rangle}$$
$$=C_d(\kappa)e^{\kappa\langle V,A\rangle}e^{\kappa\langle V,\tilde{V}-A\rangle}$$
$$=f_{\text{vMF}}(A\mid V,\kappa)\sum_{i=0}^{\infty}\frac{(\kappa\langle V,\tilde{V}-A\rangle)^i}{i!} \tag{71}$$
$$=f_{\text{vMF}}(A\mid V,\kappa)(1+\kappa\langle V,\tilde{V}-A\rangle+R_1(\tilde{V})).$$

with $R_1(\tilde{V})=\sum_{i=2}^{\infty}\frac{(\kappa\langle V,\tilde{V}-A\rangle)^i}{i!}$. Leveraging the linearity property of both integration and expectation (Jacod and Protter, 2004), we can study $P_{\text{vMF-exp}}(A\mid n,d,V,\kappa)$ by assessing separately the contribution of the different terms of the expansion of $f_{\text{vMF}}$ in:

$$P_{\text{vMF-exp}}(A\mid n,d,V,\kappa)=$$
$$\mathbb{E}_{\mathcal{X}_n\sim\mathcal{U}(\mathcal{S}^{d-1})}\left[\int_{\tilde{V}\in\mathcal{S}_{\text{Voronoï}}(A\mid\mathcal{X}_{n+1})}f_{\text{vMF}}(A\mid V,\kappa)(1+\kappa\langle V,\tilde{V}-A\rangle+R_1(\tilde{V}))\,\mathrm{d}\tilde{V}\right]. \tag{72}$$

However, contrary to the 2D case where $\mathcal{S}_{\text{Voronoï}}(A\mid\mathcal{X}_{n+1})$ is always defined as the arc between 2 angles on the circle, for $d>2$ the shape of $\mathcal{S}_{\text{Voronoï}}(A\mid\mathcal{X}_{n+1})$ is highly dependent of the layout of

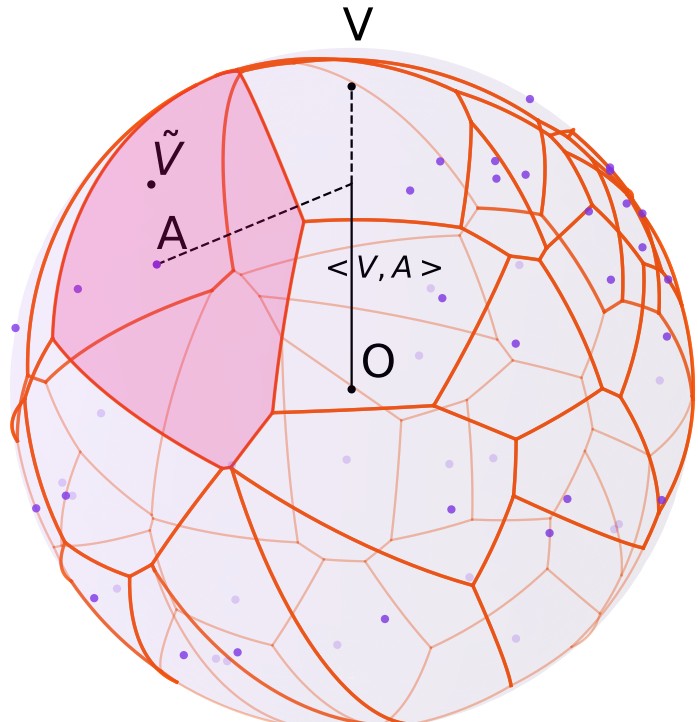

Figure 6: For $d = 3$: vMF-exp explores the action $A$ when $\tilde{V}$ lies in its Voronoï cell, shown in red.

the elements of $\mathcal{X}_n$ that share a frontier with $A$. Figure 6 provides an illustration of the complexity and diversity of the shapes of Voronoï cells for uniformly sampled points on the 3D sphere.

As a consequence, expliciting the bounds of integration, as we did in the 2D case, can be somewhat tedious. Instead, we will leverage the geometrical properties of the problem at hand to estimate $P_{\text{vMF-exp}}(A \mid n, d, V, \kappa)$. We start with the zero-order term.

### C.2.1 ZERO-ORDER TERM

Since the zero-order term is constant, its integral over $\mathcal{S}_{\text{Voronoï}}(A \mid \mathcal{X}_{n+1})$ can be expressed as:

$$\int_{\tilde{V} \in \mathcal{S}_{\text{Voronoï}}(A|\mathcal{X}_{n+1})} f_{\text{vMF}}(A \mid V, \kappa) \, \mathrm{d}\tilde{V} = f_{\text{vMF}}(A \mid V, \kappa)\mathcal{A}(\mathcal{S}_{\text{Voronoï}}(A \mid \mathcal{X}_{n+1})), \qquad (73)$$

where $\mathcal{A}(\mathcal{S}_{\text{Voronoï}}(A \mid \mathcal{X}_{n+1}))$ is the value of the surface area of $\mathcal{S}_{\text{Voronoï}}(A \mid \mathcal{X}_{n+1})$. To assess the expected value of the above equation for uniformly distributed $\mathcal{X}_n$, we use Lemma C.1 and obtain:

$$\mathbb{E}_{\mathcal{X}_n \sim \mathcal{U}(\mathcal{S}^{d-1})} \left[ \int_{\tilde{V} \in \mathcal{S}_{\text{Voronoï}}(A|\mathcal{X}_{n+1})} f_{\text{vMF}}(A \mid V, \kappa) \, \mathrm{d}\tilde{V} \right] = \frac{f_{\text{vMF}}(A \mid V, \kappa)\mathcal{A}(\mathcal{S}^{d-1})}{n+1} \qquad (74)$$

$$= \frac{f_{\text{vMF}}(A \mid V, \kappa)\mathcal{A}(\mathcal{S}^{d-1})}{n} + \mathcal{O}(\frac{1}{n^2}).$$

### C.2.2 FIRST-ORDER TERM

We want to estimate the value of:

$$\mathbb{E}_{\mathcal{X}_n \sim \mathcal{U}(\mathcal{S}^{d-1})} \left[ \int_{\tilde{V} \in \mathcal{S}_{\text{Voronoï}}(A|\mathcal{X}_{n+1})} f_{\text{vMF}}(A \mid V, \kappa)\kappa\langle V , \tilde{V} - A \rangle \, \mathrm{d}\tilde{V} \right] \qquad (75)$$

$$= f_{\text{vMF}}(A \mid V, \kappa)\kappa \left( \left\langle V , \mathbb{E}_{\mathcal{X}_n \sim \mathcal{U}(\mathcal{S}^{d-1})} \left[ \int_{\tilde{V} \in \mathcal{S}_{\text{Voronoï}}(A|\mathcal{X}_{n+1})} \tilde{V} \, \mathrm{d}\tilde{V} \right] \right\rangle - \frac{\langle V , A \rangle \mathcal{A}(\mathcal{S}^{d-1})}{n} \right).$$

Using Lemmas C.2 and C.3 as well as Corollary C.5.1, the left term inside the parentheses is:

$$\langle V \ , \ \mathbb{E}_{\ \mathcal{X}_n \sim \mathcal{U}(\mathcal{S}^{d-1})} \left[ \int_{\tilde{V} \in \mathcal{S}_{\text{Voronoï}}(A|\mathcal{X}_{n+1})} \tilde{V} \, \mathrm{d}\tilde{V} \right] \rangle$$

$$= \langle V \ , \ A \rangle \frac{\mathcal{A}(\mathcal{S}^{d-1})}{n+1} \mathbb{E}_{\ \mathcal{X}_n \sim \mathcal{U}(\mathcal{S}^{d-1}), \tilde{V} \sim \mathcal{U}(\mathcal{S}^{d-1})} \left[ \max_i \langle \tilde{V} \ , \ X_i \rangle \right] \tag{76}$$

$$= \langle V \ , \ A \rangle \frac{\mathcal{A}(\mathcal{S}^{d-1})}{n+1} \left( 1 - \frac{\Gamma(\frac{d+1}{d-1})}{2} \left( \frac{(d-1)\mathrm{B}(\frac{1}{2}, \frac{d-1}{2})}{n} \right)^{\frac{2}{d-1}} + o(\frac{1}{n^{\frac{2}{d-1}}}) \right).$$

Reinjecting this expression into Equation (75) gives the following expression for the contribution of the first-order term to the probability of sampling $A$:

$$\mathbb{E}_{\ \mathcal{X}_n \sim \mathcal{U}(\mathcal{S}^{d-1})} \left[ \int_{\tilde{V} \in \mathcal{S}_{\text{Voronoï}}(A|\mathcal{X}_{n+1})} f_{\text{vMF}}(A \mid V, \kappa) \kappa \langle V \ , \ \tilde{V} - A \rangle \, \mathrm{d}\tilde{V} \right] \tag{77}$$

$$= -f_{\text{vMF}}(A \mid V, \kappa) \frac{\mathcal{A}(\mathcal{S}^{d-1})}{n+1} \kappa \langle V \ , \ A \rangle \left( \frac{\Gamma(\frac{d+1}{d-1})}{2} \left( \frac{(d-1)\mathrm{B}(\frac{1}{2}, \frac{d-1}{2})}{n} \right)^{\frac{2}{d-1}} + o(\frac{1}{n^{\frac{2}{d-1}}}) \right).$$

### C.2.3 REMAINDER TERM

As done in the 2D proof, we leverage the Taylor-Lagrange inequality (Abramowitz and Stegun, 1948). The second derivative of the function $f(x) = C_d(\kappa)e^{\kappa x}$ is $f(x)^{(2)} = \kappa^2 f(x)$, which is bounded on $x \in [-1, 1]$ by $M = \kappa^2 C_d(\kappa)e^{\kappa x}$. This implies that:

$$|R_1(\tilde{V})| \leq \frac{M \langle V \ , \ \tilde{V} - A \rangle^2}{2}$$

$$\leq \frac{M \|\tilde{V} - A\|_2^2}{2} \text{ (according to the Cauchy-Schwarz inequality (Jacod and Protter, 2004))}$$

$$= M(1 - \langle \tilde{V} \ , \ A \rangle). \tag{78}$$

This inequality holds for every $\tilde{V} \in \mathcal{S}_{\text{Voronoï}}(A \mid \mathcal{X}_{n+1})$ when $\mathcal{X}_n \sim \mathcal{U}(\mathcal{S}^{d-1})$, which means that:

$$\mathbb{E}_{\ \mathcal{X}_n \sim \mathcal{U}(\mathcal{S}^{d-1})} \left[ \int_{\tilde{V} \in \mathcal{S}_{\text{Voronoï}}(A|\mathcal{X}_{n+1})} f_{\text{vMF}}(A \mid V, \kappa) |R_1(\tilde{V})| \, \mathrm{d}\tilde{V} \right]$$

$$\leq f_{\text{vMF}}(A \mid V, \kappa) \mathbb{E}_{\ \mathcal{X}_n \sim \mathcal{U}(\mathcal{S}^{d-1})} \left[ \int_{\tilde{V} \in \mathcal{S}_{\text{Voronoï}}(A|\mathcal{X}_{n+1})} M(1 - \langle \tilde{V} \ , \ A \rangle) \, \mathrm{d}\tilde{V} \right]$$

$$= f_{\text{vMF}}(A \mid V, \kappa) \frac{\mathcal{A}(\mathcal{S}^{d-1})}{n+1} M (1 - \mathbb{E}_{\ \mathcal{X}_{n+1} \sim \mathcal{U}(\mathcal{S}^{d-1})} \left[ \int_{\tilde{V} \in \mathcal{S}^{d-1}} \max_i \langle \tilde{V} \ , \ X_i \rangle \, \mathrm{d}\tilde{V} \right]) \tag{79}$$

$$= f_{\text{vMF}}(A \mid V, \kappa) M \frac{\mathcal{A}(\mathcal{S}^{d-1})}{n+1} \left( \frac{\Gamma(\frac{d+1}{d-1})}{2} \left( \frac{(d-1)\mathrm{B}(\frac{1}{2}, \frac{d-1}{2})}{n} \right)^{\frac{2}{d-1}} + o(\frac{1}{n^{\frac{2}{d-1}}}) \right)$$

$$= O(\frac{1}{n^{1+\frac{2}{d-1}}}).$$

We used Lemmas C.2 and C.3 to go from line 2 to 3, and Corollary C.5.1 to go from line 3 to 4. In essence, we have bounded the contribution of $R_1(\tilde{V})$ to the probability of sampling $A$ as follows:

$$\mathbb{E}_{\ \mathcal{X}_n \sim \mathcal{U}(\mathcal{S}^{d-1})} \left[ \int_{\tilde{V} \in \mathcal{S}_{\text{Voronoï}}(A|\mathcal{X}_{n+1})} f_{\text{vMF}}(A \mid V, \kappa) |R_1(\tilde{V})| \, \mathrm{d}\tilde{V} \right] = O(\frac{1}{n^{1+\frac{2}{d-1}}}) \tag{80}$$

Finally, adding up Equations (74), (78), and (80), we conclude the proof of Proposition 4.3 for $d \geq 3$ and (via the first-order term) simultaneously justify the approximate probability $P_1(a \mid n, V, \kappa)$ introduced in Proposition 4.4. $\square$

# D  SIMILAR ASYMPTOTIC BEHAVIOR OF B-EXP AND vMF-EXP FOR LARGE ACTION SETS (PROOF OF PROPOSITION 4.1)

Finally, Propositions 4.2 and 4.3 allow us to derive Proposition 4.1, i.e., that in the setting of Section 4.1, we have:

$$\lim_{n \to +\infty} \frac{P_{\text{B-exp}}(a \mid n, d, V, \kappa)}{P_{\text{vMF-exp}}(a \mid n, d, V, \kappa)} = 1. \tag{81}$$

*Proof.* Acording to Proposition 4.2, we have:

$$P_{\text{B-exp}}(a \mid n, d, V, \kappa) = \frac{f_{\text{vMF}}(A \mid V, \kappa)\mathcal{A}(\mathcal{S}^{d-1})}{n} + o(\frac{1}{n\sqrt{n}}). \tag{82}$$

Moreover, according to Proposition 4.3, we have:

$$P_{\text{vMF-exp}}(a \mid n, d, V, \kappa) = \frac{f_{\text{vMF}}(A \mid V, \kappa)\mathcal{A}(\mathcal{S}^{d-1})}{n} + \begin{cases} \mathcal{O}(\frac{1}{n^2}) & \text{if } d = 2, \\ \mathcal{O}(\frac{1}{n^{1+\frac{2}{d-1}}}) & \text{if } d > 2. \end{cases} \tag{83}$$

Therefore:

$$\begin{aligned} \lim_{n \to +\infty} \frac{P_{\text{B-exp}}(a \mid n, d, V, \kappa)}{P_{\text{vMF-exp}}(a \mid n, d, V, \kappa)} &= \lim_{n \to +\infty} \frac{n}{n} \frac{P_{\text{B-exp}}(a \mid n, d, V, \kappa)}{P_{\text{vMF-exp}}(a \mid n, d, V, \kappa)} \\ &= \frac{f_{\text{vMF}}(A \mid V, \kappa)\mathcal{A}(\mathcal{S}^{d-1}) + 0}{f_{\text{vMF}}(A \mid V, \kappa)\mathcal{A}(\mathcal{S}^{d-1}) + 0} \\ &= 1. \end{aligned} \tag{84}$$

$\square$

# E  LINK WITH THOMPSON SAMPLING

At first glance, one might draw some similarities between vMF-exp and Thompson Sampling (TS) with Gaussian prior for contextual bandits (Chapelle and Li, 2011). Admittedly, vMF-exp shares a common spirit with TS, where action selection is preceded by sampling individual weights according to a Normal distribution centered on an observed context/state vector. However, vMF-exp also presents two major differences:

- Firstly, in vMF-exp, vector sampling is performed according to a vMF hyperspherical distribution, centered on the state embedding vector $V$. This choice of distribution ensures that vectors with the same inner product with the state vector have the same probability of being sampled, as illustrated in Figure 1(a). This aligns better with the similarity used to retrieve nearest neighbors and, as emphasized in this paper, leads to probabilities of exploring actions asymptotically comparable to Boltzmann Exploration (with better scalability) under the theoretical assumptions of Section 4.1.

- Secondly, vMF-exp is not designed to maximize the expected reward of a policy in an RL or contextual bandit environment and does not impose any parameter update strategy. Instead, it serves as an action selection tool for any scenario where policy updates cannot be performed regularly (as in the batch RL setting commonly found in industrial applications), yet broad exploration must still be guaranteed between consecutive updates.

# F  MONTE CARLO SIMULATIONS

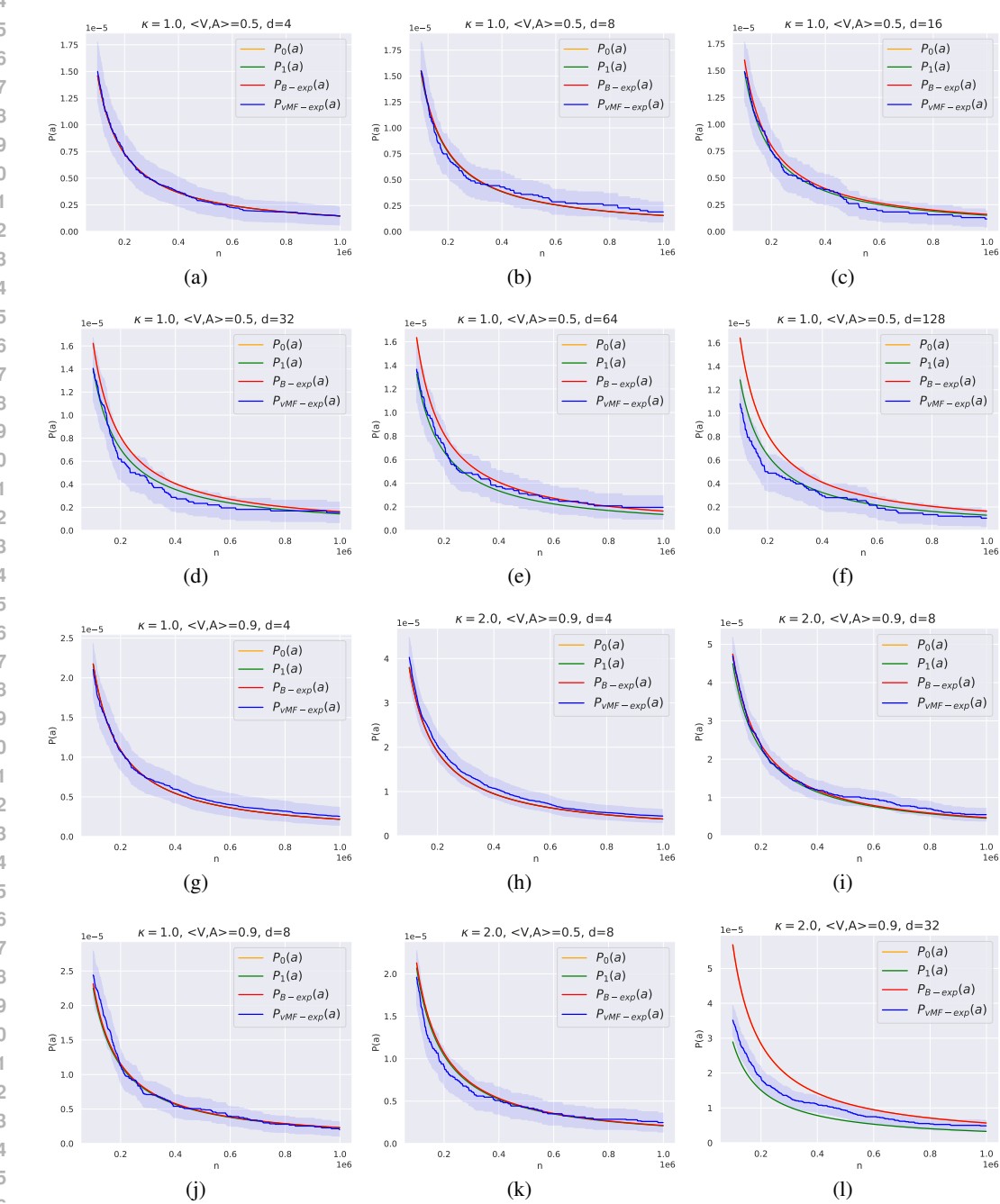

Figure 7: We report complete results for the Monte Carlo simulations presented and discussed in Section 4.3, involving more combinations of $d$, $\kappa$, and $\langle V ,\ A \rangle$). We recall that $P_{\text{B-exp}}(a)$ and $P_0(a)$ are indistinguishable for this range of $n$ values. We emphasize that the y-axis is on a 1e-5 scale; hence, all probabilities are extremely close.

# G  Sampling from the von Mises-Fisher Distribution

## G.1  Radial-tangent decomposition

Given a vector $\tilde{V} \in \mathcal{S}^{d-1}$ and a concentration $\kappa \in \mathbb{R}^+$, the algorithm described in Pinzón and Jung (2023) sample from a vMF$(V, \kappa)$ by leveraging the radial-tangent decomposition of the elements of $\mathcal{S}^{d-1}$. For any $\tilde{V} \in \mathcal{S}^{d-1}$, let's call $\tilde{t} = \langle V , \tilde{V} \rangle$. Then we have:

$$\tilde{V} = \tilde{t}V + \sqrt{1 - \tilde{t}^2}\tilde{V}_O \tag{85}$$

where the vector $\tilde{V}_O$ has a unit norm and is orthogonal to $V$.

## G.2  vMF distribution

If, $\tilde{V} \sim$ vMF$(V, \kappa)$, then :

- $\tilde{t}$ is a scalar valued random variable.
- $\tilde{V}_O$ is a random vector uniformly distributed on the (d-2) dimensional sub-sphere that is centered at and perpendicular to $V$. For instance, for $d = 3$, this would mean a circle centered around $V$.
- $\tilde{t}$ and $\tilde{V}_O$ are independent.

Since the reciprocal is also true, $t$ and $\tilde{V}_O$ can thus be separately sampled to obtain $\tilde{V}$.

## G.3  Sampling $\tilde{t} = \langle V , \tilde{V} \rangle$

The PDF of $\tilde{t}$ is known (Fisher, 1953) and follows:

$$f_{\text{radial}}(t; \kappa, d) = \frac{(\kappa/2)^{\frac{d}{2}-1}}{\Gamma(\frac{1}{2})\Gamma(\frac{d-1}{2})I_{\frac{d}{2}-1}(\kappa)}e^{t\kappa}(1 - t^2)^{\frac{d-3}{2}} \tag{86}$$

This PDF can be used to sample r through rejection sampling (Gentle, 2009).

## G.4  Sampling $\tilde{V}_O$

$\tilde{V}_O$ can be obtained by following the steps of algorithm 1.

---
**Algorithm 1:** Sample $\tilde{V}_O$

---

1 Sample vector $U$ uniformly from $\mathcal{S}^{d-1}$;
2 Compute projection of $U$ on $V$: $W = \langle U , V \rangle V$;
3 Subtract projection and normalize: $\tilde{V}_O = \frac{U-W}{||U-W||}$;
4 **return** $\tilde{V}_O$

---

Note that a simple way to sample U uniformly on $\mathcal{S}^{d-1}$ is to sample $d$ standard Gaussians independently (one for each dimension) and then normalize the resulting vector (Gentle, 2009).

## G.5  Wrapping up

The vector $\tilde{V}$ can now be computed by summing the right term of equation 86. Overall, we see that sampling $\tilde{V}$ from vMF$(V, \kappa)$ is **data-independent**, hence the scalability of the approach.

# H  ADDITIONAL EXPERIMENTS ON A PUBLIC DATASET: GLOVE-25 WORD EMBEDDINGS

While our main contributions in this work are theoretical, we aimed in the main paper to complement our key findings with experimental evaluations to illustrate the performance of vMF-exp. The main paper reported both Monte Carlo simulations, which involved reproducible synthetic data, and an online A/B test conducted on the XXX music streaming service, using real-world private data from this service. However, due to industrial constraints, we were unable to provide detailed information about the real-world data used in the A/B test or release our private dataset for reproducibility.

Understanding that some readers may wish to further explore our topic through reproducible experiments on public real-world data, we present an additional study in this section.

Specifically, in this section we report experiments comparing the behaviors of B-exp and vMF-exp for selecting tokens among the public Glove-25 [4] dataset of word embeddings (Pennington et al., 2014). All vectors represent a word token, and embeddings are learned using Word2Vec (Mikolov et al., 2013) trained on 2 billion tweets from `X.com`.

We use the public set of 1 million vectors of dimension $d = 25$, and as preprocessing we subtract to each vector the average of the entire set and divide it by its norm to ensure all vectors belong to the unit-sphere, leading to a set of unit-vectors of dimension 25, denoted $\mathcal{G}$.

The experiments follow the same protocol as the Monte Carlo estimations of Section 4.3, but instead of sampling $\mathcal{X}_n$ from the uniform spherical distribution, this time vectors are randomly sampled from $\mathcal{G}$. The vector $V$ and $A$ are also sampled from $\mathcal{G}$ so that $\langle V , A \rangle$ equals a fixed desired value. Given $\mathcal{X}_n$, we aim to assess the probability that $A$ is selected when the current state vector is $V$, using either B-exp or vMF-exp. Specifically, we want to verify whether Propositions **P1**, **P2** and **P3** defined in Section 2 hold. We also want to assess to which extent the approximations obtained in Section 4 can anticipate the behavior of B-exp and vMF-exp, despite the fact that real-world embeddings are used instead of embeddings drawn from the uniform spherical distribution.

## H.1  **P1**: SCALABILITY

As explained earlier in the paper, the efficiency of B-exp is determined by the time required to compute $\langle V , X_i \rangle$ for all $X_i \in \mathcal{X}_n$ (Section 2.2), whereas for vMF-exp it is determined by the time required to find the approximate nearest neighbor (ANN) of $\tilde{V}$ within $\mathcal{X}_n$ (Section 3.2).

We report in Table 1 the performances of popular ANN algorithms on the Glove-25 dataset, extracted from the benchmark[5] of Aumüller et al. (2017). To be consistent with the standards of ANN literature (see for instance the report of the 2023 NeurIPS competition on ANN (Simhadri et al., 2024)), the performance metric used is the maximum *throughput*, measured in number of queries-per-second (QPS), for which the average recall of the exact top-10 neighbors is greater than 90%. The algorithms chosen are two implementations of hnsw (Malkov and Yashunin, 2016), one from the library faiss (Douze et al., 2024) and the other one from the library nmslib (Boytsov and Naidan, 2013), as well as the algorithms scann (Guo et al., 2020; Sun et al., 2023) and NGT-qg (Iwasaki and Miyazaki, 2018). We also report the throughput when using exhaustive search (named "bruteforce" in Table 1) which is informative of the limited efficiency of B-exp.

Table 1 shows that performing exhaustive computations on the Glove-25 dataset for nearest neighbors search leads to a throughput 2 to 3 orders of magnitude lower than using ANN methods. This observation directly impacts the relative efficiency of B-exp compared to vMF-exp on the Glove-25 dataset, leading us to conclude that vMF-exp is scalable and verifies **P1** while B-exp does not.

Table 1: Throughput of ANN algorithms on the Glove-25 dataset. The number of queries-per-second (QPS) is increased by 2 to 3 orders of magnitude compared to bruteforce.

| Algorithm | bruteforce | hnwsm-faiss | hnsw-nmslib | scann | NGT-qg |
|-----------|:----------:|:-----------:|:-----------:|:-----:|:------:|
| QPS | 34 | 6197 | 14080 | **23436** | **22733** |

---

[4]Available at `https://nlp.stanford.edu/projects/glove/`

[5]`https://ann-benchmarks.com/glove-25-angular_10_angular.html`

## H.2 **P2,P3**: Unrestricted Radius and Order Preservation

Figure 8 reports, for $\kappa = 1$, for $20000 \leq n \leq 100000$ and for $\langle V , A \rangle \in \{0.9, 0.3, 0, -0.3, -0.9\}$, the probabilities of sampling vector $A$ when the current vector state is $V$, using either B-exp (Figure 8(a)) or vMF-exp (Figure 8(b)). Sampling is performed 30 million times to obtain a significant estimate of $P_{\text{vMF-exp}(a)}$.

Figure 8(a) illustrates the two properties that make Boltzmann exploration a popular choice for performing "soft" max on small action sets: the ability to select any action, even those with low similarity to $V$ (**P2**), and the ordering of action selection probabilities based on the similarity to $V$.

Importantly, Figure 8(b) demonstrates that our method, vMF-exp, also satisfies these two important properties. Specifically, $A$ always has a positive probability of being sampled, with this probability increasing as a function of $\langle V , A \rangle$. Thus, both B-exp and vMF-exp verify properties **P2** and **P3** on the Glove-25 dataset. Additionally, since vMF-exp is scalable (**P1**), this experiment demonstrates that vMF-exp is a relevant alternative to B-exp for discrete, very large action sets represented by embedding vectors.

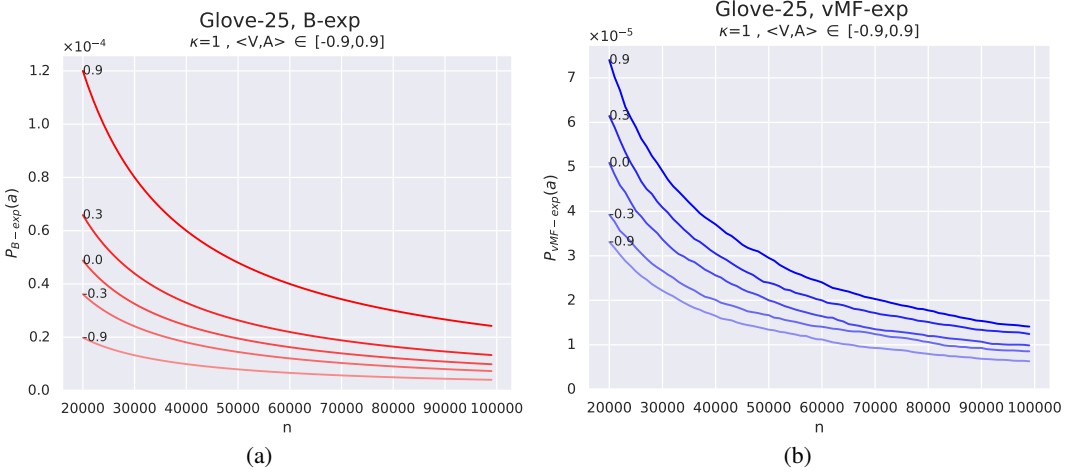

Figure 8: We report the observed probabilities of selecting vector $A$ using B-exp (a) and vMF-exp (b), on the Glove-25 dataset, for $\kappa = 1$, $20000 \leq n \leq 100000$ and $\langle V , A \rangle \in \{0.9, 0.3, 0.0, -0.3, -0.9\}$. In this regime, for both methods, the probability of selecting $A$ is strictly positive even for low values of $\langle V , A \rangle$ (**P2**) and is a strictly increasing function of $\langle V , A \rangle$ (**P3**).

## H.3 Theoretical Approximations

In this section, we compare the probabilities of selecting $A$ using B-exp and vMF-exp on the Glove-25 dataset with the probabilities predicted by the theoretical analysis in Section 4, as shown in Figure 9.

As observed in the subfigures, despite relaxing the uniform distribution assumption, the asymptotic expression from Proposition 4.2 holds across all configurations. Consequently, the yellow curve is indistinguishable from the red curve in every plot.

Also, $P_{\text{vMF-exp}}$ (blue) remains very close to $P_{\text{B-exp}}$ (red) for values of $\langle V , A \rangle$ near zero (Figures 9(a), 9(b), and 9(c)), as anticipated by Proposition 4.3 and 4.1. On Figure 9(b) and Figure 9(c), the small difference between $P_{\text{vMF-exp}}$ and $P_{\text{B-exp}}$ is anticipated by the alternate asymptotic expression (green) derived in Proposition 4.4. However, as $\langle V , A \rangle$ grows in absolute value (Figure 9(d) and Figure 9(e)), the difference between $P_{\text{vMF-exp}}$ and $P_{\text{B-exp}}$ increases faster than anticipated by Proposition 4.4, outlining the limits of the uniform distribution assumption.

Overall, the experiments show that the mathematical results obtained in Section 4 are useful to anticipate most of the respective behaviors of $P_{\text{vMF-exp}}$ and $P_{\text{B-exp}}$ on the Glove-25 dataset. To derive even more precise results and explain the behavior of vMF-exp on Figure 9(d) and Figure 9(e), future

work would benefit from modeling real-world embedding with other distributions to attempt to derive propositions equivalent to the ones presented in Section 4.

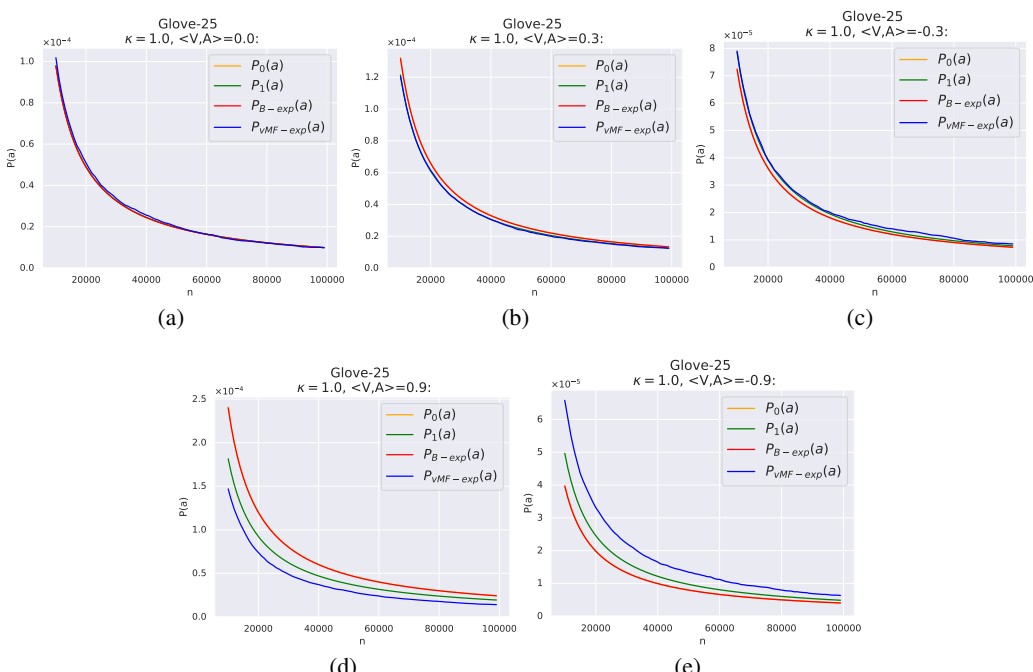

Figure 9: We compare the probabilities, on the Glove-25 dataset, of selecting $A$ using B-exp and vMF-exp with the theoretical results obtained in Section 4. Despite the relaxation of the uniform distribution assumption, the expression from Proposition 4.2 holds for every configuration. Propositions 4.3 and 4.1 are also illustrated by (a), while (b) and (c) illustrate Proposition 4.4.

.

