# OpenReview forum: "Exploring Large Action Sets with Hyperspherical Embeddings using von Mises-Fisher Sampling"
_ICLR.cc/2025/Conference — Submitted to ICLR 2025_

### Official Review · Reviewer_eaK7 · 2024-10-16

**Soundness:** 4
**Presentation:** 4
**Contribution:** 4
**Rating:** 6
**Confidence:** 2

**Summary:**

Hyperspherical embeddings are increasingly important to represent actions in a variety of settings such as recommender systems (due to various convenient theoretical properties). This paper develops scalable methods to handle large action sets of hyperspherical embedding vectors in RL problems.

Although I'll admit this paper is well outside of my expertise, it does have proven success in terms of deployment on a large music platform, demonstrating that the method is indeed scalable and performant.

**Strengths:**

-- Strong theoretical basis of the described research.
-- Real, significant contribution in what's an important space. Hyperspherical embeddings are commonly used, and this paper provides a framework for using this in RL settings that other authors will likely follow up on.
-- Actually deployed at scale with much more by the way of "proven" success than most other papers.
-- Although the paper was admittedly mostly over my head, the presentation is very clear and walks the user through the relevant details quite clearly. It's certainly possible that I missed something, but I feel that my hand was sufficiently held.

**Weaknesses:**

-- Certainly reads more like an industry paper: it is very hard to distill any real results from the experiments, or comparison to baselines etc. Statements like "this resulted in 11% more songs added to playlists than a reference cohort" obfuscates many details in order to hide the real results from readers. I understand this is a constraint when writing an industry paper, but I certainly don't like it!

-- Likewise, not much by way of real baselines, or really any experimental details.

-- Even though the presentation is good, I don't like a reader's chances of reproducing or comparing against what's written here, mainly due to the above reasons

**Questions:**

-- What are the chances of releasing code (sorry if I missed this)?

-- This paper is somewhat below the expected standards for reproducibility, even if strong in other ways. What's to stop you from, e.g., generating a synthetic data, or using a public dataset other than your own, just for the sake of reporting some real numbers (in addition to the "secret" numbers which you already report, but vaguely)? I think an ideal paper of this sort can combine both "secret" results with reproducible components, but this paper seems to fall short in that regard

---

> ### Author Response · Authors · 2024-11-22
> **Response to reviewer eaK7**
>
> Thank you for your insightful review.
>
> # Code Availability
>
> The code for reproducing the Monte Carlo simulations described in Section 4.3 was provided as a zip file in the Supplementary Material of our submission. Please let me know if you're having trouble accessing it, as in that case we may figure out with the Area Chairs an alternative way to share the code.
>
> The code randomly samples $N$ embeddings of dimension $d$, as well as a state vector $V$ and an action vector $A$ such that $<V,A>$ has the desired value. It then computes the probability of selecting action $A$ using both Boltzmann exploration and von Mises-Fisher exploration. The sampling has to be performed repeatedly (at least several times $N$, which is 1 million in Figure 2 and annex F of our paper) for the observed probability to be significant.
>
> The current code samples embeddings from a uniform spherical distribution, which is the one for which we derived theoretical guarantees that our method asymptotically behaves as Boltzmann exploration (Section 4 of the paper, with proofs spanning Appendices A to D). In the future we want to add the possibility to sample embeddings from other distributions, to assess both empirically and theoretically if a similar relationship can be found.
>
> # Additional Reproducible Experiments on a Public Dataset
>
> In response to the recurring and understandable concern expressed by reviewers that the online experiments presented in the papers can not be independently reproduced, we have performed additional experiments on a large scale public dataset of embeddings comparing the behavior of our method with the popular (but inefficient) Boltzmann sampling. We have thus submitted a revised version of our paper that includes an additional appendix (appendix H) thoroughly describing the experiments performed as well as their result. We will include the code for these experiments in our final submission.
>
> Due to the absence, to our knowledge, of large scale public datasets of embeddings for recommendations, we used instead the Glove-25 dataset [1] made of 1 million embeddings of dimension 25 representing word tokens, trained with Word2Vec on 2 billion tweets.
>
> The experiments show that, on this real-world dataset, **our method verifies the good properties described in Section 2 of our paper**. Moreover, the mathematical propositions derived in Section 4 of our paper under the simplifying assumption of uniform spherical distribution still help to anticipate most of the behavior of our method as well as Boltzmann sampling on this real-world dataset. This is especially interesting as in our opinion, **the theoretical analysis as well as the proofs provided are the main contribution** of our submission. The code for this additional experiments will be made public along the paper.
>
> # Online Experiments
>
> We understand that the manner in which results are reported in Section 5.1 aren't quantitatively informative of the performances of our method in its goal of fostering exploration in a very large discrete action space, and that it contrasts with how results on public benchmarks are usually reported.
>
> The reason we believe that Section 5 can still be valuable to the reader is two-fold:
> - It gives a concrete real-world example of an environment where the action space is made of millions of discrete actions represented by hyperspherical embeddings.
> - It shows that designing a dedicated exploration method for this problem, as opposed to using more common naive methods, does bring improvement, even though we unfortunately cannot precisely quantify it for confidentiality reasons.
>
> As such, these experiments were included to illustrate that the problem of very large action sets exploration is worth working on as it remains understudied despite its concrete applications.
>
> [1] Jeffrey Pennington, Richard Socher, and Christopher D. Manning. 2014. GloVe: Global Vectors for
> Word Representation. In Empirical Methods in Natural Language Processing (EMNLP). 1532–
> 1543.

---

### Official Review · Reviewer_oFjd · 2024-11-02

**Soundness:** 3
**Presentation:** 3
**Contribution:** 3
**Rating:** 5
**Confidence:** 2

**Summary:**

In this paper, the authors propose a new exploration strategy for reinforcement learning problems. I find the idea intriguing, but my primary concerns center on the evaluation of the proposed approach.

**Strengths:**

1. The topic of this paper is highly interesting.
2. The paper provides theoretical results.

**Weaknesses:**

1. Exploration is a key challenge in reinforcement learning, with wide-ranging applications in areas such as gaming and robotics.
2. While the paper discusses solutions for addressing high-dimensionality issues, it lacks empirical results to substantiate these claims.

**Questions:**

Sorry, I did not closely examine the mathematical derivations in this paper. I appreciate the authors’ contributions, particularly the theoretical results provided. However, my main concerns lie in the evaluation section of the paper. As the authors mention, exploration is a fundamental issue in RL. I would expect the authors to include simulations to empirically assess the exploration performance of their proposed technique. These simulations could involve game AI environments, such as those used in [1], or even synthetic data environments to provide a controlled setting for evaluation. Additionally, I strongly recommend comparing their approach with established RL exploration methods, such as count-based exploration [1] and reward-free exploration [2], as baselines.

[1] #Exploration: A Study of Count-Based Exploration for Deep Reinforcement Learning
[2] Reward-Free Exploration for Reinforcement Learning.

---

> ### Author Response · Authors · 2024-11-22
> **Response to reviewer oFjd (Part 1/2)**
>
> Thank you for your feedback and for your questions. We especially appreciate that you value our theoretical results, as we believe they constitute the most significant contribution of our paper.
> We also understand that empirical evaluation and reproducibility are essential to produce qualitative research.
>
> # Simulations on Synthetic Data
>
> Simulations on synthetic data have already been performed and are described in Section 4.3 of the paper. The corresponding code has been provided as a zip file in the supplementary material of the submission.
>
> The code randomly samples $N$ embeddings of dimension $d$, as well as a state vector $V$ and an action vector $A$ such that $<V,A>$ has the desired value. It then computes the probability of selecting action $A$ using both Boltzmann exploration and von Mises-Fisher exploration. The sampling has to be performed repeatedly (at least several times $N$, which is 1 million in Figure 2 and annex F of our paper) for the observed probability to be significant.
>
> # Experiments on a Public Dataset of Embeddings
>
> In order to improve the evaluation performed in the paper, we have performed additional experiments on a large scale public dataset of embeddings comparing the behavior of our method with the popular (but inefficient) Boltzmann sampling. We have thus submitted a revised version of our paper that includes an additional appendix (appendix H) thoroughly describing the experiments performed as well as their result. We will include the code for these experiments in our final submission.
>
> Due to the absence, to our knowledge, of large scale public datasets of embeddings for recommendation, we used instead the Glove-25 dataset [3] made of 1 million embeddings of dimension 25 representing word tokens, trained with Word2Vec on 2 billion tweets.
>
> The experiments show that, on this real-world dataset, **our method verifies the good properties described in Section 2 of our paper**. Moreover, the mathematical propositions derived in Section 4 of our paper under the simplifying assumption of uniform spherical distribution stills helps to anticipate most of the behavior of our method as well as Boltzmann sampling on this real-world dataset. This is especially interesting as in our opinion, **the theoretical analysis as well as the proofs provided are the main contribution of our submission**.
>
> [3] Jeffrey Pennington, Richard Socher, and Christopher D. Manning. 2014. GloVe: Global Vectors for
> Word Representation. In Empirical Methods in Natural Language Processing (EMNLP). 1532–
> 1543.

---

> > ### Author Response · Authors · 2024-11-22
> > **Response to reviewer oFjd (Part 2/2)**
> >
> > # Comparison to baselines
> >
> > Thank you for sharing  the papers [1] and [2] with us. Although these papers do deal with scalable exploration, they tackle a different problem than our work.
> >
> > Indeed, the scalability problem tackled in these approaches is related to **sample efficiency**; they want to find guarantees that a given exploration method will correctly assess values of states using a reasonable number of training examples.
> >
> > By contrast, the scalability problem that we approach is related to the context of **real-time decision making** when millions of actions are available; as described in Section 2.1, we want to find guarantees that an exploration method can sample any action (P2) in real-time (P1) following a distribution that favors actions based on their similarity to a given state vector (P3).
> >
> > We summarize below, in our own words, the exploration methods described in those paper.
> >
> > [1] This paper deals with exploration in environments with large state space and small action space. It proposes to scale count-based exploration, efficient in simple environments where tabular models are used, to large state space environment. It does so by hashing state representations so that states sharing the same hash also share the same visitation count that is then use to foster exploration. Their experiments are run on rllab benchmark and Arcade Environments, where the state space is made of video frames described by pixel values. The state space is thus indeed large, however the largest discrete action space used in their experiments has **18 different actions**. In the end, the stochastic policy used in this method assumes sampling from a discrete distribution where each action has a probability of being sampled that is explicitly computed, and we have explained in Section 2.2 why any such policy can not scale to environments with millions of actions.
> >
> > [2] This paper deals with exploration in environments with discrete state and action spaces. It considers the setting where a first phase of reward-free exploration is performed, during which a policy can be learned, then an arbitrary reward function is revealed. The goal is to achieve near-optimal performances on the second phase by training on the lowest possible number of examples in the first phase. The method proposed favors the visitation of "significant states" in the first phase, learning transition probabilities, then uses planning when the reward function is revealed to optimize for the second phase.
> > The paper doesn't deal with the difficulties arising when sampling from large discrete action spaces. The action selection process is based on building a Q-function with, given the current state, a value for each action that is used to sample the next action, **which doesn't scale to millions of actions**.
> >
> > Due to requiring to explicitly compute the probabilities of each action before sampling, **the methods above cannot be used as baselines for our setting**. By contrast, the method we introduce circumvents the necessity of computing explicit probabilities for each actions, as it samples a vector and then selects its approximate nearest neighbor.
> >
> > More generally, since most public benchmarks on RL focus on gaming an robotic, where the number of actions seldom exceeds one hundred, we have so far been struggling to find baseline exploration methods capable of sampling from an action set made of millions of elements in real-time to compare to.

---

### Official Review · Reviewer_rKBK · 2024-11-02

**Soundness:** 3
**Presentation:** 3
**Contribution:** 2
**Rating:** 5
**Confidence:** 3

**Summary:**

This paper introduces a tailored exploration strategy for a unique setting where hyperspherical embedding vectors represent actions, and the number of possible actions can scale to millions. The paper is well-structured and offers a robust theoretical guarantee framework. The proposed method demonstrates improved scalability compared to B-exp.

**Strengths:**

Please see the questions

**Weaknesses:**

Please see the questions

**Questions:**

1.	While the approach is interesting, the novelty seems somewhat limited. The authors incorporate hyperspherical embedding vectors into reinforcement learning, with the exploration based on a nearest-neighbor (NN) method. This approach leverages existing techniques to address a domain-specific problem, which may make the technical novelty appear modest.
2.	The method assumes that action embeddings are i.i.d. and uniformly distributed vectors. This assumption could be challenging, particularly for the recommendation scenario in this study, where actions are often interdependent. Further examination or relaxation of this assumption might enhance applicability.
3.	The motivation for employing von Mises-Fisher (vMF) exploration could be elaborated. Clarifying why this approach is particularly suitable for this setting would strengthen the rationale behind the method.
4.	The experimental setup could benefit from additional information and a broader scope:
o	Since scalability in large action spaces is a key advantage of this method, providing details on the dataset—such as size and relevant statistics—would be informative.
o	Only one unpublished dataset is used without providing sufficient details. Given that the authors claim the method as a general solution for large action spaces, validating it on benchmark datasets or RL simulations would be valuable.
o	Information on any A/B testing would add context, and including offline test results could offer additional insights.
o	Expanding the comparison methods to include recent advancements in deep reinforcement learning, which can better model large state and action spaces, would create a more comprehensive evaluation to better position the contribution of this work.
5.	It would also be helpful to include further discussion on the method’s limitations, such as challenges related to deep reinforcement learning or considerations around the exploration-exploitation trade-off in reinforcement learning.

---

> ### Author Response · Authors · 2024-11-22
> **Response to reviewer Reviewer rKBK (Part 1/2)**
>
> Thank you for your review and your questions.
>
> # Novelty of the Approach
>
> It is right that the approach builds on already existing methods of approximate nearest neighbors (ANN), however the method itself is novel as to the best of our knowledge such approach has never been proposed to perform exploration in very large discrete action spaces.
>
> More importantly, in the paper, **we prove the mathematical tie between Boltzmann exploration and von Mises-Fisher exploration** in a specific setting, and provide a non trivial proof for it (Appendices A to D). Although the problem of efficiently sampling from a softmax (i.e a Boltzmann distribution) when the number of choices is large is a notoriously difficult problem, to our knowledge this is the first time that an approach using a **continuous probability distribution**, for which sampling elements can be performed in constant time with, is proposed, with mathematically proven asymptotic properties.
>
> # Embedding Distribution Assumptions
>
> Here we would like to stress out that the uniform spherical assumption is only required to guarantee the theoretical results of Section 4. However, the method can still be employed, with scalability, **regardless of the distribution of the set of action embeddings**. To further highlight this point and to assess whether the theoretical findings apply to distributions other than the spherical uniform, we have performed some additional **experiments on a large public dataset of real-world embeddings** that we have included in the revised version of the paper (see below for details).
>
> # Motivations for von Mises-Fisher Exploration
>
> We apologize if the motivations for using von Mises-Fisher exploration were not clearly stated in the paper. We summarize below the arguments for using it in a setting where there is a very large action space, and refer to sections in the paper that develop those arguments:
>
> - Sampling from discrete distributions has a time complexity that grows linearly with the number of actions. This is problematic when there is a real-time constraint and the number of actions exceeds millions (Section 2.2).
>
> - By contrast, sampling from a continuous vector distribution has a constant time complexity with regard to the number of actions, and ANN methods have sublinear complexity, hence the two can be combined to perform **real-time sampling of discrete actions**, even when the number of actions exceeds millions (Section 3.2)
>
> - Of all continuous vector distributions, the von Mises-Fisher distribution has a probability density function that is proportional to the exponential of the dot product between the action vector and the state vector, just like the probability mass function of a Boltzmann distribution (Section 3.1). In fact, we mathematically show in the paper that if the number of actions is very large and the embeddings are uniformly distributed, Boltzmann exploration and von Mises-Fisher exploration become the same method, with the latter being scalable.

---

> > ### Author Response · Authors · 2024-11-22
> > **Response to reviewer Reviewer rKBK (Part 2/2)**
> >
> > # Experimental Setup
> >
> > We understand and agree that empirical evaluation and reproducibility are essential to produce qualitative research.
> >
> > ## Additional Experiments on a Public Dataset
> >
> > In response to the concern expressed by reviewers that the online experiments presented in the papers can not be independently reproduced, we have performed additional experiments on a large scale public dataset of embeddings comparing the behavior of our method with the popular (but inefficient) Boltzmann sampling. We have submitted a revised version of our paper that includes an additional appendix (appendix H) thoroughly describing the experiments performed as well as their result. We will include the code for these experiments in our final submission.
> >
> > Due to the absence, to our knowledge, of large scale public datasets of embeddings for recommendations, we used instead the Glove-25 dataset [1] made of 1 million embeddings of dimension 25 representing word tokens, trained with Word2Vec on 2 billion tweets.
> >
> > The experiments show that, on this real-world dataset, **our method verifies the good properties described in Section 2** of our paper. Moreover, the mathematical propositions derived in Section 4 of our paper under the simplifying assumption of uniform spherical distribution stills helps to anticipate most of the behavior of our method as well as Boltzmann sampling on this real-world dataset. This is especially interesting as in our opinion, the theoretical analysis as well as the proofs provided are the main contribution of our submission.
> >
> > ## Simulations on Synthetic Data
> >
> > Simulations on synthetic data have already been performed and are described in Section 4.3 of the paper. The corresponding code has been provided as a zip file in the supplementary material of the submission.
> >
> > The code randomly samples $N$ embeddings of dimension $d$, as well as a state vector $V$ and an action vector $A$ such that $<V,A>$ has the desired value. It then computes the probability of selecting action $A$ using both Boltzmann exploration and von Mises-Fisher exploration. The sampling has to be performed repeatedly (at least several times $N$, which is 1 million in Figure 2 and annex F of our paper) for the observed probability to be significant.
> >
> > ## Online Experiments and Comparison with Baselines
> > We apologize if Section 5.1 did not describe clearly enough the experimental setup of the online music recommendation scenario that we studied. We review below the environment considered in these experiments.
> > - The action set is made of **2 million songs**. Each of them is a represented by an embedding vector of dimension 128.
> > - At each time step, the current state $V$ is the embedding of the song that the user has listened to.
> > - The agent is expected to select the next song to recommend out of the 2 million possible songs.
> > - The policies are evaluated on both their ability to recommend songs that users add to their favorite (exploitation) and their ability to recommend novel songs given a state vector $V$ (exploration).
> >
> > Of the three policies described in section 5.1, vMF-exp obtained the best results.
> >
> > Unfortunately, since most public benchmarks on RL focus on gaming an robotic, where the number of actions seldom exceeds one hundred, we have so far been struggling to find baseline exploration methods capable of sampling from an action set made of millions of elements in real-time to compare to. If the reviewer is aware of such methods from recent advancement in deep reinforcement learning, we would greatly appreciate it if they could share them with us.
> >
> > These experiments were in fact included in our paper so as to illustrate that the problem of very large action sets exploration is worth working on as it remains understudied despite its concrete applications.
> >
> > # Discussion on the Method's Limitations
> >
> > We understand that identifying the shortcomings of the method is essential to correctly position it within the literature of exploration-exploitation trade-off. To this end, we will include the following remarks in the concluding section of our paper.
> >
> > First, the scope of the method, i.e environments with large discrete action sets where hyperspherical embeddings representing actions are available, puts a constraint on the type of applications where this method can be used.
> > Second, the theoretical analysis tying our method to Boltzmann exploration also implies that the shortcomings of Boltzmann exploration could apply to von Mises-Fisher exploration. Those would typically be:
> > - The quality of explored actions is highly dependent on the embeddings ability to represent actions based on the expected reward obtained when selecting those actions. If actions are poorly embedded, the method could favor actions with low reward.
> > - Like the temperature for Boltzmann exploration, the concentration parameter $\kappa$ determines the exploration-exploitation trade-off, making the method sensitive to the correct tuning of one hyperparameter.

---

### Official Review · Reviewer_mgSr · 2024-11-03

**Soundness:** 3
**Presentation:** 3
**Contribution:** 2
**Rating:** 5
**Confidence:** 2

**Summary:**

The authors propose to utilize the von Mises-Fisher (vMF) distribution to efficiently handle large action spaces in high-dimensional settings. The approach combines strong theoretical support with real-world validation, showcasing its effectiveness and practical applicability.

**Strengths:**

1. The authors presents a novel application of the von Mises-Fisher (vMF) distribution to efficiently explore large action spaces, improving scalability in high-dimensional environments.
2. The authors offers solid theoretical support, showing that the method achieves similar exploration efficiency to traditional approaches with lower computational costs.
3. Practical validation demonstrates the method’s relevance and potential for real-world impact

**Weaknesses:**

Novelty and Related Work: There is no section on related work to contextualize previous studies. I have noticed that some works[1-3] utilize Von Mises-Fisher or have a similar motivation to this work, especially [1,4]. Some of [1-3] employ vMF distribution to enable efficient, directed exploration in high-dimensional environments, aligning exploration toward relevant states or actions. Then by producing directionally aligned samples, vMF could find the optimal paths or guided trajectories, minimizing exhaustive search in large action spaces. I suggest the authors review this field and add more discussion of previous works to highlight this work’s unique contributions.

[1] APS: Active Pretraining with Successor Features. Hao Liu, Pieter Abbeel Proceedings of the 38th International Conference on Machine Learning, PMLR 139:6736-6747, 2021.

[2] Mecanna, Selim, Aurore Loisy, and Christophe Eloy. "Applying Reinforcement Learning to Navigation In Partially Observable Flows." Seventeenth European Workshop on Reinforcement Learning.

[3] Guo X, Chang S, Yu M, et al. Faster Reinforcement Learning with Expert State Sequences[J]. 2018.

[4] Zhu, Yiwen, et al. "vMFER: Von Mises-Fisher Experience Resampling Based on Uncertainty of Gradient Directions for Policy Improvement." arXiv preprint arXiv:2405.08638 (2024).

**Questions:**

Refer to the weaknesses.

---

> ### Author Response · Authors · 2024-11-16
> **Response to Reviewer mgSr**
>
> Thank you for your review.
>
> It is correct that no section is explicitly named "related work" in our submission, however we listed the limitations of the (few) existing solutions to perform exploration in very large discrete action sets with millions of elements in Section 2.2 of our submission, and cited the paper [4] in Section 2.3 when introducing the von Mises-Fisher (vMF) distribution.
>
> We appreciate that you took the time to share several publications leveraging the vMF distribution in a RL context. However, to our understanding, these papers do not tackle the problem of exploration when actions space is discrete and very large. Below, we provide a brief summary, in our own words, of each of these publications.
>
> [1] This paper introduces an unsupervised pretraining objective for RL where the agent is allowed to interact with the environment without being informed of the reward obtained, and seeks to satisfy 2 objectives:
> - exploring the most "diverse" set of states, mathematically characterized by representing each state $s$ by a vector $\phi(s)$ and ensuring that the distribution of states visited by the policy has maximal entropy.
> - ensuring that given a task, defined by a vector $w$, the set of states explored is relevant to this task. A state $s$ is considered relevant to the task if $<w,\phi(s)>$ is high, which is equivalent to saying that $s$ has a high log-likelihood of being sampled from vMF$(w, 1)$.
>
> The paper does not deal with the problem of sampling actions when the action set is very large; indeed, on line 8 of Algorithm 1, it is stated that they use $\epsilon$-greedy to sample actions during the data collection phase, and we have explained in Section 2.2 of our paper why it is not possible when the action set is very large (millions of actions), whereas their experiments are run on the suite of Atari games where __the maximum number of actions is 18__.
>
> [2] This papers studies the problem of helping an agent navigate in an environment that is either a 2D or 3D flow with unknown dynamics. During episodes, the agent is pushed forward by a constant velocity vector, and the goal at each time step is to decide the orientation of the agent. Thus, the problem is modeled as a POMPD where, unlike our setting, __the action space is continuous__ and gives the orientation for the agent.
> Performances of Q-learning, A2C and PPO are compared. For A2C and PPO, action is sampled from a vMF distribution centered on a vector predicted by a neural-network actor, which makes more sense than the more common Gaussian distribution because the action to be selected represents a direction instead of a position, but this paper does not deal with the problem of sampling from large discrete action spaces.
>
> [3] This papers studies how to imitate expert behavior when successive states of expert trajectories are available, but actual actions taken are unknown. Their method represents states in a latent space, and the parameters of the function representing states are learned by performing gradient ascent on an expected cumulated reward. To compute this gradient, the distribution of the shift between two subsequent states is assumed to be a vMF centered on the current sub-goal representation (defined in the paper).
> Here __vMF is not used to sample actions__;  it is said in Section 3.1 that when running the agent in an environment to augment dataset, the action to be performed at each time step is selected __greedily__. This is possible because the experiments are run on a suite of 5 environments related to the Doom game for which the maximum number of available actions is 7, contrasting with our settings where actions range in millions.
>
> [4] This paper deals with the issue of using the gradients of an ensemble of critics when improving a policy. As the critics may highly disagree on the value of taking an action given a state, their respective gradients may also disagree about the direction towards which a parameter update should be made. The authors propose to assign during training a value of gradient uncertainty to each transition of the replay buffer, and sample transitions based on the level of certainty on the gradient's direction. This certainty is computed by taking, for each transition, the set of gradient vectors of the ensemble of critics, and fitting a vMF distribution using the same method we refer to in our online experiments of Section 5.1. The concentration parameter $kappa$ of the distribution is then used as certainty.
> The experiments that they describe are performed on Mujoco robotic control environments, where __action spaces are continuous__ and multi-dimensional, and so they do not deal with sampling from large discrete action spaces.
>
> All of the above articles share the common property of dealing with data where direction matters more than position, and so does our contribution, but none of them tackles the problem of sampling from extremely large discrete action sets, which is an active problem.

---

> > ### Author Response · Authors · 2024-11-18
> >
> > To summarize the above answer:
> >
> > We intend to revise Section 2.3 of our submission, where the vMF distribution is introduced, so as to mention [1], [2] and [3] as additional examples of usage of the vMF distribution in an RL context, although on problems different from the one tackled in our paper, i.e very large action space exploration, thus highlighting the unique contributions of our paper in contrast to previous work.

---

### Meta-Review · Area_Chair_ETH5 · 2024-12-21

**Metareview:**

The paper received four reviews with ratings of 5, 5, 5, and 6. It applies approximate nearest neighbor exploration and von Mises-Fisher sampling to improve exploration in large discrete action spaces. The problem considered is important and the authors present reasonable ideas for how to address this challenge. However, the paper requires substantial revision before it can be considered for publication. The biggest weakness identified by the reviewers is the limitation of the evaluation, which was judged to be inadequate from a reproducibility perspective, lacking baselines, and not fully substantiating the practical claims of the paper.  Another significant criticism is that the assumptions for the theoretical analysis are overly restrictive.  A number of prior works were overlooked, and there was no section on related work to contextualize the contribution. Based on the majority of reviewers' opinions, this paper is recommended for rejection.

**Additional Comments On Reviewer Discussion:**

Although there was no direct discussion with the authors, the overall opinions of the reviewers converge towards rejection.

---

### Decision · Program_Chairs · 2025-01-22

Reject